# 1-Bit Wonder: Improving QAT Performance in the Low-Bit Regime through K-Means Quantization

**Sohir Maskey** [* 1]  **Constantin Eichenberg** [* 1]  **Johannes Messner** [* 1]  **Douglas Orr** [* 2]

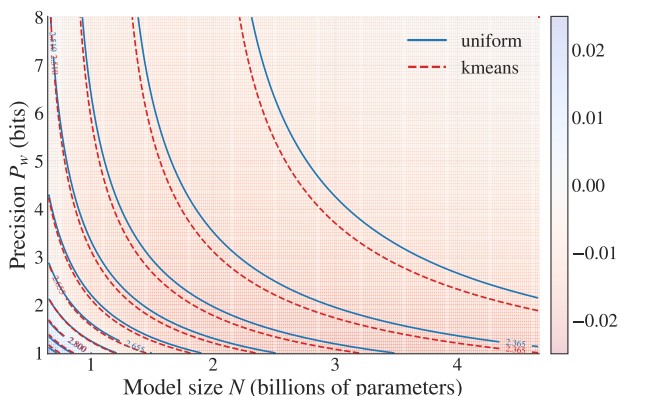
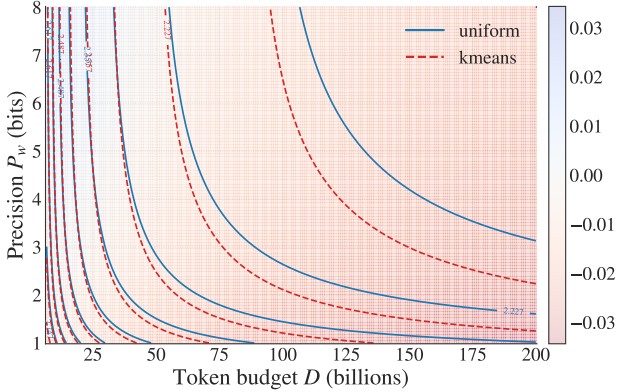

*Figure 1.* **IsoLoss contours under precision–budget tradeoffs.** (Left) Precision–parameter tradeoff under a fixed 50B token budget. (Right) Precision–token tradeoff under a fixed 3.9B parameter budget. Background colors show the predicted loss gap between uniform integer quantization and k-means quantization (red indicates lower loss for k-means). Across all evaluated regimes, k-means strictly dominates uniform quantization, highlighting the consistent advantage of nonlinear formats under fixed memory budgets.

## Abstract

Quantization-aware training (QAT) is an effective method to drastically reduce the memory footprint of LLMs while keeping performance degradation at an acceptable level. However, the optimal choice of quantization format and bit-width presents a challenge in practice. The full design space of quantization is not fully explored in the context of QAT, and the precise trade-off between quantization and downstream performance is poorly understood, as comparisons often rely solely on perplexity-based evaluations. In this work, we address these shortcomings with an empirical study of QAT in the low-bit regime. We show that k-means based weight quantization outperforms integer formats and can be implemented efficiently on standard hardware. Furthermore, we find that, under a fixed inference memory budget, the best performance on generative downstream tasks is achieved with 1-bit quantized weights.

---
[1] Aleph Alpha Research [2] Graphcore Research. Correspondence to: Sohir Maskey <sohir.maskey@aleph-alpha-research.com>.

*Proceedings of the $43^{rd}$ International Conference on Machine Learning*, Seoul, South Korea. PMLR 306, 2026. Copyright 2026 by the author(s).

## 1. Introduction

Large language models (LLMs) have demonstrated remarkable performance across a wide range of language tasks, largely driven by scaling model size (Brown et al., 2020; Chowdhery et al., 2022; Wei et al., 2022). However, this increase in parameter count comes at the cost of substantial memory footprint and bandwidth demands during both training and inference (Shoeybi et al., 2019; Bommasani et al., 2021). For moderate batch sizes, inference is typically memory-bound rather than compute-bound (Frantar et al., 2024; Mazurek & Gabriel, 2025), making memory bandwidth and weight storage performance bottlenecks.

Recent work demonstrates that reducing weight precision can substantially alleviate inference bottlenecks in LLMs. For instance, Frantar et al. (2024) show that 4-bit weight quantization enables near-ideal speedups for autoregressive LLM inference by mitigating memory bandwidth constraints. Motivated by such results, weight quantization has become a widely adopted approach for reducing inference-time costs, and is employed in frontier LLMs such as Kimi-K2 (Moonshot AI, 2025). By representing model weights in lower precision, quantization reduces both the memory footprint of the model and the volume of data transferred between memory and compute units, enabling practical speedups of memory-bound inference.

These memory reductions and speedups come with an inherent trade-off: quantization introduces approximation error that can adversely affect model performance. However, the extent to which accuracy degrades, particularly at very low bit-widths, remains an open and actively debated question. On the one hand, several works argue that quantization-aware training enables aggressive low-bit quantization with little to no loss in accuracy (Wang et al., 2023; Ma et al., 2024). On the other hand, empirical studies report that substantial performance degradation when reducing weight precision below 6 bits (Kumar et al., 2025). This discrepancy highlights the need for a more systematic and principled understanding of weight quantization.

To understand the source of these conflicting findings, a closer look at prior work reveals that much of the debate around aggressive low-bit quantization is driven by differences in evaluation methodology. Several influential works advocating ternary or 1-bit weights primarily rely on training or validation loss, or token-level log-likelihood evaluations, to assess model quality (Wang et al., 2023; Ma et al., 2024). At the same time, studies reporting severe degradation at low bit-widths often restrict their analysis to similar loss-based metrics, without consistently incorporating state-of-the-art quantization techniques such as block scaling factors or nonlinear quantizers (Kumar et al., 2025). As a result, it remains unclear to what extent observed performance gaps reflect fundamental limitations of aggressive quantization, as opposed to artifacts of the evaluation protocol or quantization design choices. In particular, evaluations beyond loss are necessary, since recent work reports substantial degradation on more realistic downstream generative tasks even under moderate quantization (Li et al., 2024).

**Contributions** In this work, we revisit a central question underlying efficient LLM inference: *given a fixed inference memory budget, how should one trade off parameter count and weight precision?* To this end, we contribute the following results:

1. We conduct a precision-aware scaling law analysis with two findings:
   
   (a) Using fewer bits is generally preferable under a fixed weight memory budget.
   
   (b) A format based on scalar k-means quantization outperforms the commonly used symmetric integer format at low bit-widths.

2. We complement our analysis with generative downstream evaluations and fair, memory-matched comparisons, pairing a 4B parameter `bf16` model trained on 150B tokens with its 4-bit and 1-bit counterparts of approximately 12B and 31B parameters, respectively.

3. We demonstrate that efficient inference for the k-means

format is feasible on standard hardware using vector lookup tables.

Furthermore, we release our training code, the kernels, and the model checkpoints for our 4B, 12B, and 31B models, together with the corresponding inference code.

## 2. Method

This section presents our methodology, beginning with background on scalar quantization and quantization-aware training (QAT) in Section 2.1, followed by a description of the specific quantization formats and QAT schedules used in our experiments in Section 2.2.

### 2.1. Background on Quantization

**Scalar quantization** A *quantization format* is specified by the components needed to quantize real-valued weight tensors. Concretely, for the scope of this work we define a format as a tuple $(\lambda, f)$: a *scale function* $\lambda \colon \mathbb{R}^n \to \mathbb{R}_+$ and a *quantizer* $f : \mathbb{R} \to \mathbb{R}$. The quantizer $f$ is a piecewise constant function

$$
f(x) = \begin{cases} q_0 & \text{if } x < x_0, \\ q_i & \text{if } x \in [x_i, x_{i+1}), \\ q_{N-1} & \text{if } x \geq x_N. \end{cases}
$$

where $I = [x_0, x_N]$ defines an interval outside of which values get clipped to boundary bins $q_0$ or $q_{N-1}$, respectively. Because weight quantization formats are usually symmetric, we also require that $x_N = -x_0$.

Given a quantization format $(\lambda, f)$, the quantization and dequantization operations $Q$ and $\bar{Q}$ on a weight tensor $w \in \mathbb{R}^n$ are defined as

$$
Q(w) = f(w/\lambda(w)), \quad \bar{Q}(w) = Q(w) \cdot \lambda(w),
$$

where $f$ is applied entrywise. In practice, $\lambda$ is some global statistic of $w$ chosen such that the majority of entries of $w$ get mapped to $I$, e.g., $\lambda \propto \text{absmax}$.

The reconstructed tensor $\bar{Q}(w)$ is an approximation of the original tensor $w$. To materialize $\bar{Q}(w)$ we need to store both $Q(w)$ and $\lambda(w)$. Since entries of $Q(w)$ are by definition contained in the set $\{q_0, ..., q_{N-1}\}$, we can store $Q(w)$ in $n = \log_2 N$ bits per entry, which is called the *bit-width* of the format.

**Rounding to nearest centroids** A special case of the quantizer $f$ is given by

$$
f_c(x) = \arg\min_{c_i} |x - c_i|, \tag{1}
$$

for some set of centroid points $c = \{c_i\} \subset I$, with $i = 0, ..., N - 1$. In other words, $f_c$ is the function that rounds the input $x$ to its nearest neighbor among the centroids. Panter & Dite (1951) show that for a given probability distribution $p$ on $I$ and fixed number of quantization levels $N$, the piecewise constant minimizer of the reconstruction error $\mathbb{E}_p[(x - f(x))^2]$ is indeed of the form $f_c$ for some $c$. In general there is no closed form expression for the optimal centroids, but they can be obtained algorithmically by 1D k-means clustering (Lloyd, 1982).

**Block formats**  Instead of applying quantize/dequantize operations to a whole tensor $w$ we can also split $w$ into disjoint blocks $w_J$ of $B$ elements each, and apply the operations separately on each $w_J$. This means we must also store all $\lambda(w_J)$ instead of only $\lambda(w)$, slightly increasing memory overhead at the benefit of generally more accurate quantization. If $n$ bits are used to store each $q_i$ and $m$ bits are used to store $\lambda(w_J)$, the average bit allocation $P_w$ for a block quantized tensor with $d$ elements and block size $B$ is

$$P_w = \frac{1}{d}\left(d \cdot n + \frac{d}{B} \cdot m\right) = n + \frac{m}{B}. \qquad (2)$$

In this sense, block wise quantization can be interpreted as using a fractional bit-width, e.g., a 4 bit format with block size 64 where each block scale is stored as a 16 bit number has an average bit-width of 4.25 bits per element. Hence we extend our notion of format to triples $(\lambda, f, B)$, with bit-width given by Equation (2).

**Linear formats**  The most commonly used quantization formats are linear (also called uniform) formats. These are formats with a centroid quantizer $f_c$ where $c$ is a uniformly spaced grid. This has the benefit that tensor contractions (typically matrix multiplications) between $\bar{Q}(w)$ and $\bar{Q}(v)$ can be reduced to operations on integer-like types, which are often natively supported by hardware. We describe concrete examples in Section 2.2.

**Quantization-Aware Training**  Quantization-aware training (QAT) simulates quantized weights (and, optionally, activations) during training by using $\bar{Q}(w)$ in the forward pass and the unquantized weights $w$ as a surrogate for gradient computation in the backward pass. Optimizing the training objective under quantization noise typically leads to improved performance compared to applying quantization only after training (Or et al., 2024). In practice, simulated quantization is implemented by replacing

$$w \leftarrow w + \text{SG}(\bar{Q}(w) - w), \qquad (3)$$

where SG denotes the stop-gradient operation. Unlike with low-precision training, which promises training as well as inference speedups, the backward pass is executed in high pre-

cision. This means that QAT typically incurs additional compute and memory overhead compared to standard training. However, QAT is generally more stable than low-precision training and still yields speed and memory improvements at inference time, since $\bar{Q}(w)$ is sufficient to reproduce the forward pass and can be represented in fewer bits than $w$.

## 2.2. Quantization Formats and QAT Schedule

In this section, we describe the concrete quantization formats and training procedures used throughout our experiments. All schemes are instances of the general quantization framework introduced in Section 2.1, and differ only in the choice of quantizer $f$ and scale function $\lambda$. For all formats, we use the same block size $B = 64$ and store $\lambda$ in 16 bits.

**Uniform integer quantization**  Our baseline format is uniform integer quantization. The $n$-bit integer format uses a centroid quantizer $f_c$ as defined in Equation (1) and a scale function $\lambda$ defined by:

$$c = \begin{cases} \{-(2^{n-1} - 1), \ldots, 2^{n-1} - 1\} \subset \mathbb{Z}, \ n \geq 2, \\ \{-1, 1\}, \ n = 1, \end{cases}$$

$$\lambda(w) = \begin{cases} \frac{\text{absmax}(w)}{(2^{n-1} - 1)}, \ n > 2, \\ \text{absmean}(w), \ n \leq 2, \end{cases}$$

i.e., for formats of $\leq 2$ bits, we switch from absmax to absmean scaling, otherwise most values would be rounded to zero, degrading performance. For the 1-bit format, we observe severe optimization instabilities that frequently lead to NaNs early in training. To mitigate this, we follow the normalization strategy introduced in BitNet (Wang et al., 2023) and subtract the tensor-wise mean before quantization.

Regarding the bit-width of integer formats, we note that the number of centroids for $n > 1$ is $2^n - 1$. Hence, according to our definition, the bit-width is $P_w = \log_2(2^n - 1)$ which is approximated by $n$ for large $n$, but significantly differs from $n$ in the low bit regime. In particular, at $n = 2$ we have $c = \{-1, 0, 1\}$ which effectively gives ternary weights at a bit-width of $P_w \approx 1.58$. Whether to view the actual bit-width as $\log_2(2^n - 1)$ or $n$ depends on the perspective. From a theoretical point of view the former is more natural, while from a practical viewpoint one typically uses $n$ bits to store the quantized representations, since the logic to unpack non-bit-aligned values can be slow. For our scaling laws we argue it is more accurate to use $P_w = \log_2(2^n - 1)$, as this corresponds to the most compact representation possible. Following Equation (2), the average bit-width should also account for storing $\lambda$, using 16 bits per block of 64 values, an overhead of 0.25 bits per weight. The average bit-width for our $n$-bit integer formats is therefore:

| $n$ | 2 | 3 | 4 | 5 | 6 | 7 | 8 |
|-----|------|------|------|------|------|------|------|
| $P_w$ | 1.83 | 3.06 | 4.16 | 5.22 | 6.23 | 7.24 | 8.24 |

For simplicity, we still refer to these formats as the $n$-bit uniform format, despite the subtleties previously discussed.

**Nonlinear k-means quantization**  Our second quantization format is based on nonlinear centroid quantization using 1D k-means clustering. For each weight tensor, we compute a set of centroids

$$c = \{c_0, \dots, c_{N-1}\} \subset [-1, 1]$$

by running k-means on the normalized weights, and define the quantizer as in Equation (1). This is motivated by the discussion in Section 2.1, as this choice of $f_c$ minimizes the $L^2$ reconstruction error for a fixed number of quantization levels under mild assumptions (Lloyd, 1982).

As with uniform quantization, we use block-wise formats and store one scale parameter per block, again incurring an overhead of 0.25 bits per weight. The key difference lies in the choice of $c$: instead of a uniform grid, the centroids are learned and adapt to the empirical weight distribution[1]. Importantly, we find that this nonlinear format remains stable even in the 1-bit regime, without requiring mean-shifting or other stabilization tricks.

We refer to this format as the $n$-bit k-means format.

**Quantization-aware training schedule**  For all experiments, we employ QAT as described in Section 2.1. Rather than enabling QAT from the very start of training, we first perform a warm-up phase of 1 000 training steps using standard bf16 mixed precision. This design choice is motivated by two considerations. First, early training is characterized by a highly non-stationary and "chaotic" phase in which weight distributions evolve rapidly. Applying quantization during this phase can lead to unstable optimization and, in the case of k-means quantization, to trivial centroid configurations that do not reflect meaningful weight structure. Second, recent work has shown that the timing of when QAT is introduced can have a substantial impact on final model performance (Liu et al., 2025). To isolate the effect of the quantization format itself, we therefore keep the QAT start point fixed across all experiments.

After step 1 000, quantization remains active for the rest of training. For k-means quantization, the centroids are learned at the onset of QAT and then frozen. This allows QAT to adapt to a fixed set of centroids, improving training stability.

---

[1]Storing the set of centroids also increases the average bit-width. However, since $N \ll \text{numel}(w)$, this contribution is negligible.

## 3. Experimental Results

In this section, we present our experimental results for QAT. In Section 3.1, we derive scaling laws that explicitly account for the precision at which models are trained, including an analysis in weight-memory-matched settings. In Section 3.2, we address our main research question by evaluating memory-matched models: a 4B bf16 baseline and its 12B 4-bit and 31B 1-bit counterparts. These models are trained in longer runs, followed by an SFT phase, and evaluated on downstream generative benchmarks. Finally, Section 3.3 provides benchmarks of specialized kernels designed for efficient inference with the k-means format.

All models are trained with backbone-only weight quantization; embeddings and output projections remain in bf16. Additional details on the training setup, data mixture, hardware, architecture, and further ablations are provided in Section A.

### 3.1. Scaling Laws

**Experimental setup**  We train and evaluate a suite of language models on Nemotron-CC (Su et al., 2025), using a standard Llama-style dense Transformer architecture (Grattafiori et al., 2024).

Our experimental grid spans model size, training data, precision, and quantization format. Specifically, we consider parameter counts

$$N \in \{0.8, 1.4, 3.9\} \text{ (billions)},$$

training token budgets

$$D \in \{8.4, 16.8, 25.2, 33.6, 41.9, 50.3\} \text{ (billions)},$$

and weight precisions

$$P_w \in \{1.25, 2.25, 3.25, 4.25, 6.25, 8.25\}$$

for the k-means format, and

$$P_w \in \{1.25, 1.83, 3.06, 4.16, 6.23, 8.24\}$$

for the uniform format.

All models are trained from scratch following the QAT schedule described in Section 2.2. Evaluation is performed using pretraining loss, which provides an unbiased estimate of test loss in the infinite-data regime (training tokens $\ll$ total corpus size).

**Scaling law model with low precision parameters**  Following Kaplan et al. (2020); Hoffmann et al. (2022), we adopt the standard decomposition of language model loss into three terms: one relating to model capacity, another to the amount of training data $D$ and a final irreducible loss

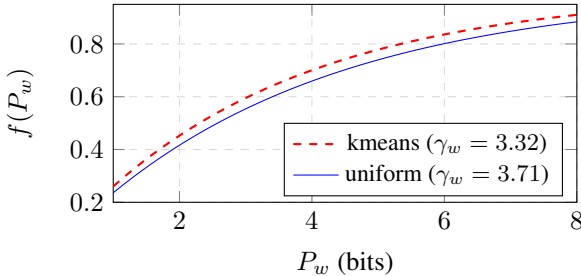 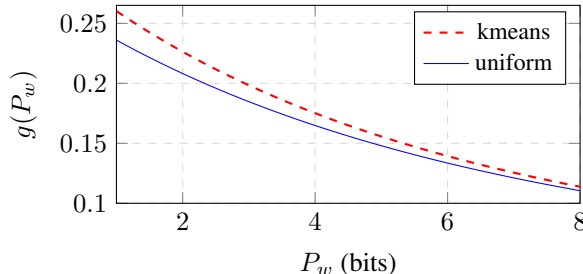

*Figure 2.* Precision-to-capacity mapping and the induced memory-normalized efficiency. (Left) The saturating function $f(P_w)$ models diminishing returns as weight precision increases, where higher values are better. (Right) The ratio $g(P_w) = f(P_w)/P_w$ determines the optimal precision under a fixed inference-memory budget $M = N P_w$, where higher values are better.

term $E$. However, instead of treating the raw parameter count $N$ as the sole proxy for model capacity, we explicitly incorporate weight precision by introducing a *effective parameter count* $N_{\text{eff}}$, as suggested by Kumar et al. (2025). Concretely, we model the loss as

$$\mathcal{L}(N, D, P_w) = A N_{\text{eff}}(N, P_w)^{-\alpha} + B D^{-\beta} + E, \quad (4)$$

where $A, B, E, \alpha, \beta > 0$ are fitted constants.

To capture the diminishing returns of increasing bit-width, we map the weight precision $P_w$ to effective capacity using a saturating function

$$f(P_w) = 1 - \exp\left(-\frac{P_w}{\gamma_w}\right), \quad (5)$$

with a learned scale parameter $\gamma_w > 0$. The effective parameter count is then defined as $N_{\text{eff}}(N, P_w) = N f(P_w)$. This formulation ensures that reducing precision lowers the effective capacity of the model, while recovering the classical scaling law behavior as $P_w$ increases and $f(P_w) \to 1$.

**Fit quality and parameter estimates** All parameters are estimated via L-BFGS, see Section B.1 for more details. Both quantization formats are well described by the proposed scaling law, achieving $R^2 \approx 0.96$ and low RMSE. Figure 6 plots predicted versus observed loss. However, the fitted parameters reveal systematic differences between linear and nonlinear quantization.

For k-means quantization, we obtain

$$\alpha \approx 0.63, \qquad \beta \approx 0.40, \qquad \gamma_w \approx 3.32,$$

indicating a strong dependence of loss on effective model capacity (large $\alpha$) and a relatively fast saturation of capacity with increasing bit-width (small $\gamma_w$).

In contrast, uniform quantization yields

$$\alpha \approx 0.55, \qquad \beta \approx 0.46, \qquad \gamma_w \approx 3.71,$$

corresponding to weaker scaling with model size and a slower saturation in precision. We plot the predictions of these fits in Figure 1.

> **Key takeaway.** K-means quantization achieves uniformly lower loss than uniform integer quantization across all precision–budget tradeoffs, with the largest improvements in the ultra-low-bit regime.

**Optimal precision under a memory budget** A central motivation of this work is to understand how to allocate a *fixed inference memory budget* between model size and weight precision. Assuming the dominant cost is weight storage, the memory budget can be approximated by

$$M = N P_w, \quad (6)$$

where $N$ is the parameter count and $P_w$ the bits per weight.

Under the constraint in Equation (6), we can rewrite $N = M/P_w$ and obtain $N_{\text{eff}}(M, P_w) = \frac{M}{P_w} f(P_w)$. Holding the training budget $D$ fixed, the only dependence on $P_w$ is through $N_{\text{eff}}(M, P_w)$, and minimizing the loss in Equation (4) is therefore equivalent to maximizing

$$g(P_w) := \frac{f(P_w)}{P_w}, \quad (7)$$

which can be interpreted as *effective capacity per memory bit*. This corresponds exactly to the quantity visualized in Figure 2 (right), while Figure 2 (left) illustrates the saturating behavior of $f(P_w)$.

Across both quantization formats (uniform and k-means), the ratio $g(P_w) = f(P_w)/P_w$ decreases as $P_w$ increases, indicating that under a fixed memory budget the scaling law favors allocating memory to *more parameters* rather than *more bits per parameter*. In other words, the predicted optimum is to push toward the lowest precision and reinvest the saved memory into scaling up $N$.

At the same time, the nonlinear k-means format yields consistently larger values of $g(P_w)$ in the low-bit regime, implying higher effective capacity per memory bit than uniform quantization. This advantage is most pronounced at $P_w \leq 4.25$, where nonlinear formats strictly dominate uniform ones under memory-matched comparisons.

> **Key takeaway.** Under fixed memory, the best regime is the lowest stable precision balanced by scaling up parameters. K-means formats yield higher effective capacity per bit than uniform formats.

## 3.2. Memory-Matched Scaling and Downstream Generative Performance

To understand the trade-off between model scale and weight precision under a fixed memory budget, we conduct longer training runs and evaluate the resulting models on a broad suite of downstream generative benchmarks. Unlike loss- or perplexity-based analyses, our goal is to directly assess how quantization choices affect *generative performance*.

**Training setup and data mixture** We train a 4B-parameter model in `bf16` precision on 150B tokens. Following recent best practices for strong open-weight LLMs, we adopt a curriculum inspired by Bakouch et al. (2025), in which general web data is progressively mixed with higher-quality sources such as code and math early in training. This data mixture is designed to improve reasoning and structured generation capabilities while preserving general language modeling performance. For general web and code data, we use filtered versions of Nemotron-CC (Su et al., 2025) and Starcoder (Lozhkov et al., 2024), respectively. For high-quality mathematical data, we incorporate a mixture of FineMath-3+ and FineMath-4+ (Allal et al., 2025).

To enable chat-style and instruction-following behavior, we further perform a short supervised fine-tuning (SFT) phase using the Tulu 3 SFT Mixture (Lambert et al., 2024).

**Memory-matched model variants** The 4B `bf16` baseline requires approximately 7.8GB of memory for model weights at inference time. To study how this memory budget can be reallocated between parameter count and precision, we train two additional models whose total weight memory closely matches this budget: i) a 12B-parameter model with 4-bit weights, and ii) a 31B-parameter model with 1-bit weights. See Section A.3 for exact architectures. Both models use QAT with nonlinear k-means quantization format as described in Section 2.2. Aside from model size and weight precision, training recipes and data mixtures are kept equivalent.

**Evaluation protocol** We evaluate all models on a diverse set of benchmarks covering commonsense reasoning, multiple-choice question answering, code generation, instruction following, and mathematical reasoning. These include log prob evals (MMLU (Hendrycks et al., 2021), HellaSwag (Zellers et al., 2019), PIQA (Seo et al., 2018), ARC (Clark et al., 2018)), and further generative evaluations, i.e., HumanEval (Chen et al., 2021), MBPP (Austin

*Table 1.* Memory-matched scaling evaluation results across model sizes and bit-widths.

| Benchmark | 4B/16-bit | 12B/4-bit | 31B/1-bit |
|---|---|---|---|
| MMLU | 33.21 | 50.86 | **51.61** |
| HellaSwag | 50.16 | **55.41** | 54.70 |
| PIQA | 74.97 | 76.88 | **77.20** |
| ARC-C | 41.21 | 49.15 | **50.68** |
| ARC-E | 71.34 | 75.25 | **77.27** |
| MBPP | 19.00 | 26.40 | **30.00** |
| HumanEval | 12.80 | 15.24 | **18.29** |
| HumanEval Instruct | 26.22 | 35.98 | **42.68** |
| GSM8K (5-shot) | 27.90 | **48.52** | 45.26 |
| IFEVAL | 51.16 | **63.45** | 62.70 |
| MMLU PRO COT | 9.86 | 17.00 | **17.71** |
| AidanBench | 88.08 | 147.48 | **167.83** |

et al., 2021), GSM8K (Cobbe et al., 2021), MMLU-PRO (Wang et al., 2024), and AidanBench (McLaughlin et al., 2024). We follow standard evaluation protocols and report accuracy or pass@1 as appropriate, except for AidanBench, where we report a coherence score computed by GPT-4-mini as a judge.

**Results** Table 1 reports the downstream performance of the three memory-matched models. Both low-bit models outperform the 4B `bf16` baseline across all benchmarks.

The 31B 1-bit model achieves the strongest overall results on most knowledge, reasoning, and code-generation tasks, including MMLU, ARC-Challenge, MBPP, HumanEval, and MMLU-PRO. The 12B 4-bit model remains competitive and even performs best on GSM8K.

Overall, these results validate the scaling-law prediction from Section 3.1: under a fixed inference memory budget, allocating capacity to *more parameters* rather than *more bits per parameter* yields stronger downstream generative performance. They also demonstrate that learned nonlinear quantization formats enable even 1-bit models to scale effectively without catastrophic degradation.

## 3.3. Efficiency

Our model evaluations consider a fixed inference memory budget. This choice enables comparison across formats and models without reference to a particular hardware implementation or software stack. While model size during inference can be the limiting factor in some deployment scenarios, especially with the increased prevalence of mixture-of-expert (MoE) models, inference speed and energy consumption often dominate practical considerations. Accordingly, we evaluate the practical efficiency of nonlinear 4-bit and ultra-low-precision 1-bit weight formats with block scaling.

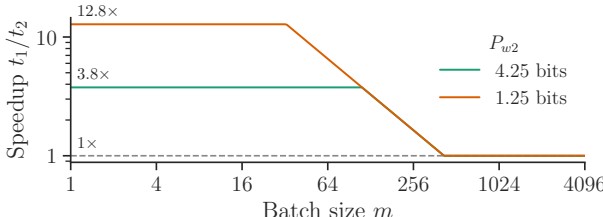

*Figure 3.* A theoretical model for speedup of 1-bit and 4-bit formats versus `bf16`, when decoded to `bf16` in software on an L40S GPU.

*Table 2.* Micro-benchmark results for the product of a $1 \times 8192$ `bf16` vector and $8192 \times 8192$ quantized matrix. The 4-bit format achieves near-optimal speedup. Peak memory bandwidth for the test GPU is $864\,\text{GB/s}$. All standard errors are $<0.2\,\mu\text{s}$.

| $P_w$ | Time (Speedup) | Effective BW |
|---|---|---|
| 16 | $175.6\,\mu\text{s}\ (1.0\times)$ | $764\,\text{GB/s}$ |
| 4.25 | $49.5\,\mu\text{s}\ (3.7\times)$ | $721\,\text{GB/s}$ |
| 1.25 | $24.0\,\mu\text{s}\ (7.6\times)$ | $438\,\text{GB/s}$ |

**Hardware formats**   We distinguish between quantization formats with native hardware support and those requiring software implementation using generic hardware capabilities. Formats with hardware support are constrained by the fundamental capabilities of the process and technology. However introducing new formats is expensive and the development cycle is long. We consider hardware formats to be out of scope for this work, and refer the reader to Rouhani et al. (2023); Sharma et al. (2018) for an evaluation of related formats. At a high level, we note that data movement is often a substantial contributor to energy use (Horowitz, 2014), such that our memory budget constraint can serve as a coarse proxy for energy cost.

**Software formats (theoretical)**   We consider an indicative theoretical model for the speedup from software-supported formats that share a common computational format. In this model, the kernel execution time is the maximum of the compute and memory transfer times, each executed at a device-dependent peak rate. The resulting speedup under some mild assumptions (see Section D) is

$$\text{speedup}_{1\to 2} = \frac{t_1}{t_2} = \frac{\max(1, P_{w1} \cdot \nu/(16 \cdot m))}{\max(1, P_{w2} \cdot \nu/(16 \cdot m))}, \quad (8)$$

where $t_1$ and $t_2$ denote execution times, $m$ is the batch size and $\nu := \text{R}_\text{compute}/\text{R}_\text{transfer}$ is the device-specific ratio of compute rate (op/s) to memory transfer rate (bytes/s).

This relation highlights three regimes. For small $m$, the $\text{speedup}_{1\to 2}$ is the ratio of the precisions $P_{w1}/P_{w2}$, as both kernels are *memory-bound* with execution time determined by the amount of data transferred from memory. For large $m$, the speedup is 1 (no speedup), as both kernels become *compute-bound* and perform approximately the same amount of arithmetic work. In the intermediate regime, the speedup scales $\propto 1/m$. For example, using `bf16` compute on the L40S GPU (NVIDIA, 2024), this model predicts a peak speedup of $16/4.25 = 3.8\times$ for 4-bit formats at batch size $m \leq 111$, with no speedup expected for $m > 418$. Similarly, 1-bit formats achieve a peak speedup of $16/1.25 = 12.8\times$ when $m \leq 32$, again with no speedup when $m > 418$ (see Figure 3).

**Software formats (practical)**   We evaluate the speed of fused dequantize-multiply operations on an inference-focused GPU, within micro-benchmarks and whole-model generation benchmarks (see Section E for full details). Table 2 shows micro-benchmark results for a setting representative of batch size 1 autoregressive inference. Our custom Triton (Tillet et al., 2019) kernel uses a 256-entry lookup table to support element formats of $\{1, 2, 4, 8\}$ bits. It achieves near-theoretical-optimum speedups for 4-bit non-linear block-scaled formats, and a substantial speedup for 1-bit formats relative to 16 and 4-bit baselines.

Additional results for other dimensions, as well as a modified Marlin 4-bit kernel (Frantar et al., 2024), which outperforms the Triton kernels for larger batch sizes, are provided in Section E. These show that, as expected, the advantage of weight quantization shrinks as batch size increases, with no speedup observed by batch size 256, although the memory capacity benefit remains.

For whole-model token generation benchmarks, we adapt the Llama 3 model from Hugging Face by replacing all backbone linear layers with adapters that call the fused dequantize-multiply kernels evaluated above. We generate 100 tokens from an empty KV cache and measure wall-clock time. To avoid measuring kernel launch overhead and host synchronization, we capture and replay the generation of a single token using a CUDAGraph, yielding high memory bandwidth utilization for the baseline.

Results are shown in Table 3, demonstrating that the 12B 4-bit model from Section 3.2 runs only slightly slower than the 4B 16-bit baseline. The 31B 1-bit model is somewhat slower, due in part to the increased kernel overhead observed in Table 2 and the extra compute from larger activations (e.g. RMSNorm and activation functions). While the larger 1-bit models do not match 4-bit throughput in this setting, they retain a substantial memory capacity advantage, enabling larger models under the same inference memory budget.

## 4. Related Work

**Post-training quantization (PTQ)**   The most common approach to reduce the average weight precision of LLMs is PTQ, since it avoids expensive retraining of the full model.

*Table 3.* Token generation performance for the models of Section 3.2 (blue), for single-user autoregressive generation. These correspond to effective bandwidth of 620, 524, and 372 GB/s for 4B 16-bit, 12B 4-bit, and 31B 1-bit models respectively. The maximum error bar ($\pm 2$ standard errors) is $< 1\%$ of the mean.

| | Tokens/s | | |
| $P_w$ | 4B | 12B | 31B |
| --- | --- | --- | --- |
| 16 | 88.8 | 30.0 | OOM |
| 4.25 | 190.1 | 79.3 | 35.4 |
| 1.25 | 245.6 | 120.0 | 60.7 |

However, simple uniform integer PTQ often degrades downstream performance, see, e.g., (Kim et al., 2024). To mitigate this, a line of work develops *sensitivity-aware* PTQ methods that explicitly account for which weights are most critical under quantization.

For example, OBQ (Frantar et al., 2022b) casts layer-wise quantization as a quadratic error minimization problem and performs greedy weight quantization with error-compensating updates based on the inverse Hessian as sensitivity measure. Building on this foundation, GPTQ-style methods (Frantar et al., 2022a; 2023) scale these second-order ideas to LLMs by using efficient blockwise updates.

AWQ (Lin et al., 2024) estimates sensitivity through activation magnitudes and selectively preserves particularly important weights in bf16. Going beyond uniform grids, SqueezeLLM (Kim et al., 2024) combines Hessian-based sensitivity estimates with non-uniform quantization. More broadly, recent work has emphasized that the choice of quantization format itself is a fundamental design axis: Orr et al. (2025) systematically study optimal scalar quantization formats and show that nonlinear centroid-based representations can significantly improve reconstruction error compared to standard uniform grids. Another complementary approach is SmoothQuant (Xiao et al., 2023), which enables joint weight and activation quantization to int8 by smoothing activation outliers and migrating quantization difficulty from activations to weights via an equivalent transformation.

Overall, while modern PTQ methods can remain competitive down to ~4-bit weights, their performance degrades rapidly below this regime, where quantization noise can no longer be absorbed without adapting model parameters. Moreover, even moderate quantization below ~8 bits has been shown to substantially harm downstream generative abilities, particularly instruction following, self-calibration, and in-context learning (Li et al., 2024).

**QAT and ultra-low-bit models**  To enable aggressive quantization many approaches rely on QAT. Representative examples include 1.58-bit (Ma et al., 2024) and 1-bit weight networks (Wang et al., 2023), which demonstrate that, with appropriate training recipes (e.g., normalization), extremely low-bit weights can remain viable. Despite this

progress, frontier deployments have historically remained conservative (often 16-bit), with a recent trend towards int4 block formats (Moonshot AI, 2025).

**Scaling laws with precision constraints**  Scaling laws (Kaplan et al., 2020; Hoffmann et al., 2022) are a standard tool to extrapolate from low-resource experiments to regime-level decisions under fixed constraints (e.g., compute- or memory-budgets), typically modeling performance as a function of model size and data. Most work treats parameter count as the primary proxy for capacity; an exception is Kumar et al. (2025), which introduces precision-aware scaling law analyses and reports optima around ~6 bits under their assumptions and evaluation protocol. More recently, Liu et al. (2025) study scaling with aggressively low precision down to 1-bit, focusing primarily on linear quantization and additionally analyzing *when* to introduce QAT during pre-training, finding that late-stage QAT can be advantageous.

**Efficient inference with nonlinear formats**  Without native hardware support, practical speedups from low-bit nonlinear weight formats rely on custom kernels that fuse dequantization with matrix multiplication, keeping weights compressed in memory and expanding them only on chip. Early lookup table (LUT)-based methods such as SqueezeLLM (Kim et al., 2024) demonstrated throughput gains from centroid-based quantization but fell short of ideal memory-bound scaling, with more recent methods such as FLUTE (Guo et al., 2024) employing offline weight restructuring and optimized LUT handling to approach bandwidth-limited performance for 4-bit matmuls at small batch sizes.

## 5. Discussion

**Limitations**  Our work advocates for nonlinear block-quantized and ultra-low-precision formats, based on a scaling laws analysis of popular formats. However, the space of formats is vast and the cost of scaling laws experiments prohibits a broad exploration. Importantly, there are many design decisions that might interact with this conclusion, such as the choice of learning rate and schedule during QAT. In terms of putting our findings into practice, our primary comparison in terms of memory capacity may be sufficient for some applications. However where the constraint is inference speed on current hardware, we observe speedups only for small batch sizes, which would not be efficient in many deployment scenarios, including those with substantial prefill compute.

**Conclusion**  Our work demonstrates the strength of nonlinear formats, with substantial advantages over uniform quantization, especially below 4.25 bits/parameter. We also demonstrate the effectiveness of ultra-low-precision formats

using block scaling, with a 31B parameter model with 1.25 bits/parameter outperforming a 12B parameter model with 4.25 bits/parameter across a wide variety of tasks. Our hope is that these results will guide and inspire further work into ultra-low-precision formats and nonlinear format design.

## Acknowledgments

We thank Callum McLean and Luka Ribar for helpful discussions and support throughout the development of this work.

## Impact Statement

This paper presents work whose goal is to advance the field of machine learning. There are many potential societal consequences of our work, none of which we feel must be specifically highlighted here.

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

*Table 4.* Pretraining curriculum mixture over data sources. Training runs for 36K steps with 4 194 304 tokens per step ($\approx$ 151B tokens total). Phase lengths follow a 72%/18%/10% split of total training. Each entry reports both mixture proportion and the corresponding token count.

| | Nemotron | | FineMath-3+ | | FineMath-4+ | | Starcoder-V2 | |
|---|---|---|---|---|---|---|---|---|
| Phase | % | Tokens | % | Tokens | % | Tokens | % | Tokens |
| Phase 1 (72%) | 85% | 92.4B | 3% | 3.3B | 0% | 0.0B | 12% | 13.0B |
| Phase 2 (18%) | 75% | 20.4B | 0% | 0.0B | 16% | 4.4B | 9% | 2.4B |
| Phase 3 (10%) | 63% | 9.5B | 0% | 0.0B | 27.4% | 4.1B | 9.6% | 1.4B |

# A. Experimental Details

All experiments were implemented in PyTorch (Paszke et al., 2019) using the `torchtitan` framework (Liang et al., 2025). All models are trained from scratch, following the QAT schedule described in the main paper. Our training code is publicly available at: `https://github.com/Aleph-Alpha-Research/1-Bit-Wonder`.

## A.1. Training Setup

**Hardware** All training runs were performed on 64 NVIDIA H100 GPUs, each with 80GB of DRAM. We used Fully Sharded Data Parallelism (FSDP2). For the 12B and 31B models, we additionally used activation checkpointing to reduce memory usage.

**Pretraining Configuration** All experiments use a global batch size of 4 194 304 tokens per optimization step, with a context length of 4 096. We use the standard Llama 3 tokenizer with vocabulary size 128 256, and append `EOS` tokens at the end of documents.

## A.2. Pretraining Data

For all pretraining runs, we fix the data sampling seed to ensure that models trained for the same number of steps deterministically see the same tokens at each optimization step.

**Scaling-law experiments** All scaling-law runs in Section 3.1 are trained on a filtered subset of Nemotron-CC (Su et al., 2025).

**Long-horizon pretraining curriculum** For the long-horizon runs in Section 3.2, we adopt a curriculum inspired by Bakouch et al. (2025), in which general web data is progressively mixed with higher-quality sources such as code and mathematics. Concretely, we combine Nemotron with filtered Starcoder-V2 data (Lozhkov et al., 2024) and FineMath-3+/4+ data (Allal et al., 2025). The exact mixture schedule is summarized in Table 4.

## A.3. Model architectures

All models in this paper follow the Llama 3 architecture with hyperparameters scaled to match the target parameter count. Concretely, we consider a standard decoder-only Transformer (Vaswani et al., 2017) trained autoregressively with causal self-attention. The model consists of a stack of Transformer blocks, each comprising (i) pre-normalization using RMSNorm, (ii) multi-head masked self-attention with RoPE (Su et al., 2021) applied to queries and keys, and (iii) a position-wise feed-forward network using a SwiGLU gating nonlinearity (Shazeer, 2020). Following the Llama 3 design, the attention module employs grouped query attention (Ainslie et al., 2023), i.e., a larger number of query heads than key/value heads, to improve computational efficiency while maintaining modeling capacity. Residual connections are used around both the attention and feed-forward sublayers.

Consistent with the Llama 3 design, we use no dropout and no bias terms in linear projections. Training is performed with a standard causal language modeling mask, ensuring that each token can only attend to past context. In addition, we apply document-level masking to prevent attention across document boundaries when processing concatenated sequences. Our implementation follows the modular reference provided in the `torchtitan` codebase for Llama 3 models.[2] Finally,

---

[2]`https://github.com/pytorch/torchtitan/tree/main/torchtitan/models/llama3`

*Table 5.* Architectural and optimization hyperparameters for all model scales. All models use SwiGLU with multiplier 1.3, RMSNorm pre-normalization, grouped query attention, FlexAttention-based SDPA, no dropout or bias terms, and gradient clipping at norm 1.0.

| Model | Parameters N | Dim $d$ | Layers $L$ | Heads | KV Heads | Max LR | RoPE $\theta$ |
|---|---|---|---|---|---|---|---|
| 0.8B | 818 714 112 | 1 536 | 12 | 12 | 6 | $8 \cdot 10^{-4}$ | 500 000 |
| 1.4B | 1 431 373 824 | 2 048 | 16 | 16 | 8 | $6 \cdot 10^{-4}$ | 500 000 |
| 3.9/4B | 3 883 551 744 | 3 072 | 24 | 24 | 8 | $4 \cdot 10^{-4}$ | 500 000 |
| 12B | 11 520 053 248 | 4 096 | 48 | 32 | 8 | $3 \cdot 10^{-4}$ | 500 000 |
| 31B | 30 643 279 872 | 6 144 | 60 | 48 | 8 | $2 \cdot 10^{-4}$ | 500 000 |

we implement scaled dot-product attention (SDPA) using PyTorch's FlexAttention operator for efficient fused attention computation.[3]

**Model scales**  We scale model depth and width from the Llama 3 8B configuration to satisfy a fixed inference weight-memory budget. Our reference point is a 4B-parameter `bf16` model, corresponding to approximately 7.8GB of weight storage (more precisely, 3 883 551 744 parameters × 2 bytes; see the third row of Table 5 for the exact architecture).

For the memory-matched low-bit variants, we quantize only the Transformer *backbone* weights, while keeping the input embedding and output projection in `bf16`. For the 31B 1.25-bit model (see the last row of Table 5), we use $d = 6 144$ and 48 attention heads, satisfying standard divisibility constraints ($d$ is a multiple of 256, $d$ is divisible by the number of heads, and the number of heads is a multiple of the number of KV heads, i.e., 8). With untied embeddings, the embedding and output projection each contain $128 256 \times 6 144 \approx 0.79$B parameters, for 1.58B parameters in `bf16` ($\approx 3.15$GB). The remaining backbone contains 29.07B parameters, which at 1.25 bits/weight requires $\approx 4.54$GB. In total, this yields $\approx 7.7$GB, matching approximately the reference memory budget. Similarly, the 12B 4.25-bit model uses $d = 4096$ with $L = 48$ layers, for a total of 11.5B parameters. Here, the backbone contains 10.47B parameters, occupying $\approx 5.56$GB at 4.25 bits/weight, while the untied embedding and output projection contribute $2 \times (128 256 \times 4 096) \approx 1.05$B parameters in `bf16` ($\approx 2.10$GB). This again matches approximately the reference memory budget.

Similarly, the 12B 4.25-bit model uses $d = 4096$ with $L = 48$ layers, for a total of 11.5B parameters. Here, the Transformer backbone contains 10.5B parameters, occupying roughly 5.6GB at 4.25 bits, while the embedding and output layers contribute about 1.0B parameters ($\sim 2.1$GB in `bf16`). This again matches the reference memory budget. See Table 5 for a detailed overview.

All models use grouped query attention with a fixed number of key/value heads ($n_{\mathrm{kv}} = 8$ for models $\geq 1$B), RoPE with $\theta = 500 000$, and SwiGLU feed-forward blocks with multiplier 1.3, with intermediate dimensions rounded to multiples of 1024. Architectural hyperparameters for all scales are summarized in

## A.4. Optimization Hyperparameters

Across all runs, we optimized models using AdamW (Kingma & Ba, 2014) with weight decay 0.1, and otherwise standard AdamW hyperparameters ($\beta_1 = 0.9$, $\beta_2 = 0.95$, $\epsilon = 10^{-8}$). We employed a learning rate schedule consisting of a warmup phase over the first 100 optimization steps, followed by a constant learning rate with a linear decay over the final 10% of training. We apply gradient clipping with a global norm of 1.0 to stabilize optimization at larger scales.

We experiment with maximum learning rates in the range $\{8 \cdot 10^{-4}, 6 \cdot 10^{-4}, 4 \cdot 10^{-4}, 3 \cdot 10^{-4}, 2 \cdot 10^{-4}\}$.

## A.5. Supervised Fine-Tuning (SFT)

After pretraining, all memory-matched models from Section 3.2 (i.e., 4B `bf16`, 12B 4-bit and 30B 1-bit models) undergo a short supervised fine-tuning phase on the Tulu 3 SFT mixture (Lambert et al., 2024). We use the Llama 3 instruction chat template together with the Llama 3 tokenizer. We train for five epochs, using the same optimizer settings as in pretraining but reducing the learning rate by a factor of $10^{-1}$. Moreover, we extend the sequence length to $8, 192$ tokens and apply standard document packing to accelerate training.

---

[3] `https://github.com/meta-pytorch/attention-gym`

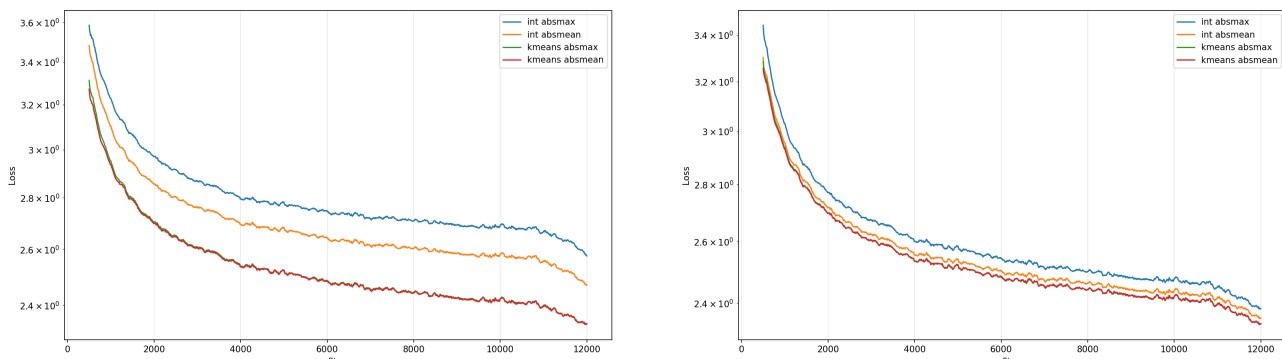

*Figure 4.* **Normalization ablation.** Training loss curves with logarithmic $y$-axis for different block-wise normalization strategies. *Left:* 2-bit quantization, where absmean scaling leads to substantially more stable optimization. *Right:* 4-bit quantization, where absmax scaling becomes favorable. K-means quantization remains largely insensitive to this choice.

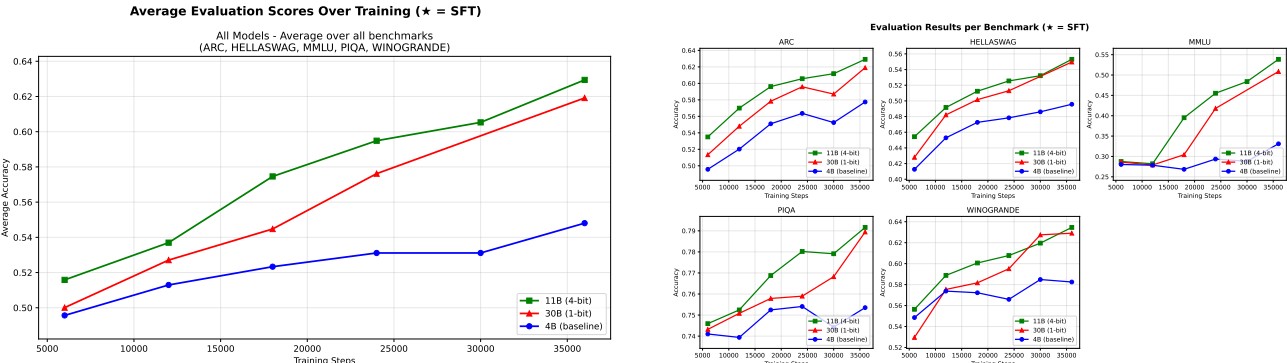

*Figure 5.* **Training progression** Evolution of pretraining evaluations during the training of the 4B `bf16`, 12B 4-bit, and 31B 1-bit models.

### A.6. Additional Ablations

We ablate the choice of block-wise normalization used in our quantization operators, comparing absmean scaling (normalization by the mean absolute value within each block) against absmax scaling (normalization by the maximum absolute value). This design choice has a substantial impact on uniform linear quantization at very low bit-widths. In particular, for extremely aggressive quantization ($\leq 2$ bits), absmean yields significantly lower training loss, whereas for moderate precision ($\geq 3$ bits), absmax becomes preferable.

In contrast, the k-means quantizer is largely insensitive to the normalization choice, exhibiting similar training loss trajectories across all tested bit-widths. Figure 4 summarizes these findings, showing the training loss trajectories (log-scale) for 2-bit and 4-bit quantization.

### A.7. Evolution of evaluation metrics

We perform logprob-based evaluations during the pretraining phase of our 4B 16-bit, 12B 4-bit, and 31B 1-bit models (Figure 5). We can see that here the 12B 4bit model is consistently on-par with or better than the 31B 1-bit model. We therefore note that in our experiments these pretraining metrics are not an accurate predictor for downstream generative model performance, where the 31B 1-bit model is on-par with or better than the 12B-4bit model (see Table 1).

## B. Scaling Laws

Scaling laws characterize how the predictive performance of LLMs improves as fundamental resources—most notably model size, data, and compute—are increased. A central empirical finding is that, across wide regimes, test loss decreases in a smooth and predictable manner, enabling reliable extrapolation and principled trade-offs between scaling dimensions. In particular, Kaplan et al. (2020) and Hoffmann et al. (2022) report that loss follows approximate power laws in model and

dataset size. A standard two-factor parametrization is

$$\mathcal{L}(N, D) \ = \ A\, N^{-\alpha} \ + \ B\, D^{-\beta} \ + \ E, \tag{9}$$

where $\mathcal{L}(N, D)$ denotes the (test) loss for a model with $N$ parameters trained on $D$ tokens, and $A, B, E, \alpha, \beta > 0$ are fitted constants. The exponents $\alpha$ and $\beta$ quantify diminishing returns: increasing $N$ or $D$ yields monotonically smaller, yet persistent, improvements.

**Precision-aware scaling**   Recent work has extended Equation (9) to account for quantization effects by interpreting reduced numerical precision as a decrease in *effective* model capacity. For example, Kumar et al. (2025) model QAT by replacing $N$ with a precision-dependent effective size $N\,f(P_w)$, where $P_w$ denotes the average weight precision. In their formulation, $f(\cdot)$ is a saturating function in bits, such that capacity approaches the full-precision regime as $P_w$ increases, i.e., $f(P_w) \approx 1$ for sufficiently large $P_w$.

**Final precision-aware functional form**   In our setting, models are trained with weights quantized to an average precision $P_w$ (including block-scale overhead). We retain the standard decomposition into a model-size term and a data-size term, but replace the raw parameter count by an effective parameter count that depends on precision:

$$\mathcal{L}(N, D, P_w) \ = \ A\, N_{\text{eff}}(N, P_w)^{-\alpha} \ + \ B\, D^{-\beta} \ + \ E. \tag{10}$$

To capture diminishing returns in bit-width, we map precision to effective capacity via a saturating function

$$f(P_w) \ = \ 1 - \exp\!\left(-\frac{P_w}{\gamma_w}\right), \tag{11}$$

with fitted scale parameter $\gamma_w > 0$. The effective parameter count is then defined as

$$N_{\text{eff}}(N, P_w) \ = \ N\, f(P_w). \tag{12}$$

This construction ensures that reducing precision lowers effective capacity, while recovering the classical scaling law behavior as $P_w$ increases and $f(P_w) \to 1$.

We empirically estimate $\mathcal{L}(N, D, P_w)$ using losses from models ranging from 0.8B to 3.9B parameters, trained on datasets containing between 8B and 50B tokens.

### B.1. Fitting Procedure

To estimate the parameters of the precision-aware scaling law in Equation (10), we follow the robust parametric fitting methodology of Hoffmann et al. (2022). As in their work, we fit the scaling law in the log domain and optimize a robust regression objective to mitigate the influence of outliers and heteroscedastic noise in empirical loss measurements.

Specifically, we reparametrize the positive coefficients $A, B, E$ as $a = \log A$, $b = \log B$, and $e = \log E$, and evaluate predicted log loss via a log-sum-exp combination of the model and data terms:

$$\log \hat{\mathcal{L}}(N, D, P_w) = \log \sum_{\text{exp}} \Big(a - \alpha \log N_{\text{eff}}(N, P_w), \ b - \beta \log D, \ e\Big), \tag{13}$$

which ensures numerical stability and preserves the additive structure of the original law.

Following Hoffmann et al. (2022), the fitting objective is a Huber loss on the log-residuals,

$$\min_{a, b, e, \alpha, \beta, \gamma_w} \sum_i \text{Huber}_\delta\Big(\log \hat{\mathcal{L}}(N_i, D_i, P_{w,i}) - \log \mathcal{L}_i\Big), \tag{14}$$

where $\mathcal{L}_i$ denotes the observed loss for the $i$-th run and $\delta = 10^{-3}$ is chosen to balance sensitivity and robustness. The Huber loss blends squared and absolute residual penalties, making the fit less sensitive to extreme observations while still maintaining high fidelity to the bulk of the data.

We enforce positivity of $\alpha, \beta, \gamma_w$ via a softplus reparameterization and optimize all parameters with the L-BFGS algorithm. To mitigate sensitivity to initialization and local minima, we perform a multi-start grid search over plausible initial values for $(a, b, e, \alpha, \beta)$, selecting the solution with the lowest robust objective.

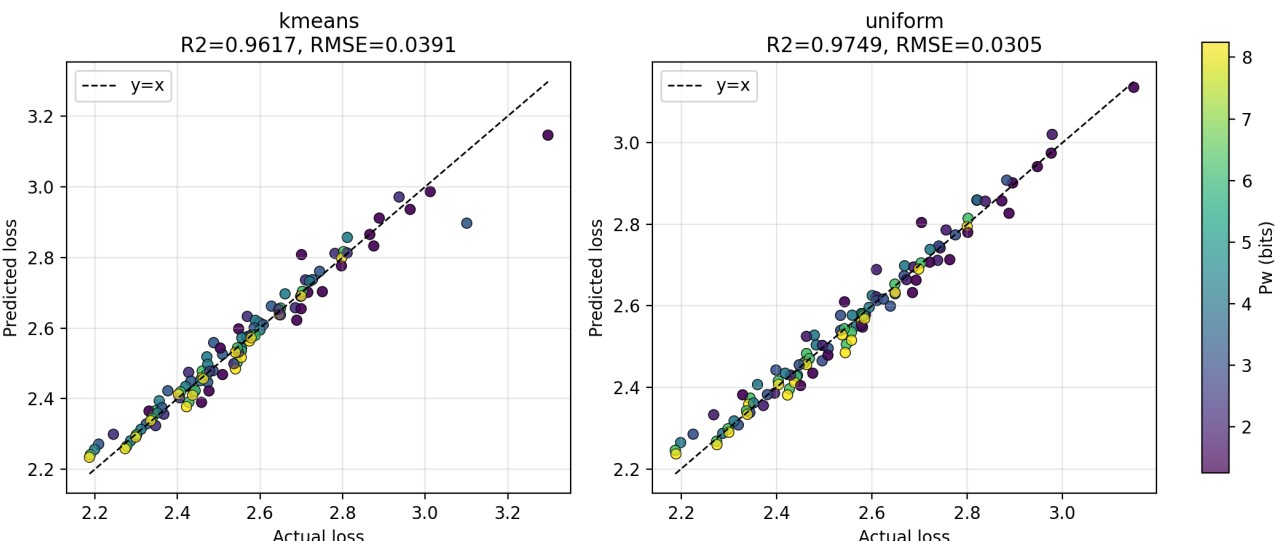

*Figure 6.* Predicted vs. actual loss for k-means (left) and uniform (right) quantization formats. Points are colored by precision $P_w$ (bits). Both methods achieve strong fits with $R^2 > 0.96$, with uniform quantization showing slightly better predictive accuracy (RMSE $= 0.0305$) compared to k-means (RMSE $= 0.0391$). The dashed line indicates perfect prediction ($y = x$).

**Fit Quality**   Both quantization formats are well described by the proposed precision-aware scaling law. Across all runs, the fitted model achieves high predictive accuracy, with $R^2 > 0.96$ and low RMSE. Figure 6 compares predicted versus observed losses, showing that the scaling law captures the systematic dependence on model size, token budget, and weight precision. While uniform quantization yields slightly lower RMSE, the overall fit quality is comparable for both formats, supporting the use of the learned saturation parameter $\gamma_w$ as a meaningful summary of precision-induced capacity loss.

## C. Extended Discussion on Scaling Laws

This section provides an extended discussion of Section 3.1. While the main body of the paper focuses on the regime in which backbone parameters dominate the memory footprint, we now consider a more general setting. In particular, we present a comprehensive analysis of how to allocate a fixed inference memory budget between model size and weight precision.

**Goal.**   We consider the common deployment setting where the dominant constraint is weight memory, i.e., only $M$ Gb (Gigabit) are available to store parameters. Lowering the backbone weight precision $P_w$ allows us to store more parameters, but extremely low precision yields diminishing returns. To capture this, we remember that we use the saturating quality factor

$$f(P_w) = 1 - \exp(-P_w/\gamma_w), \qquad \gamma_w > 0,$$

and define the effective parameter count

$$N_{\text{eff}}(P_w, N) = f(P_w) N,$$

where $N$ denotes the total number of parameters.

**Memory accounting and the induced coupling $N(P_w)$.**   We store the embedding parameters in 16 bits, while all remaining backbone parameters are stored in $P_w$ bits. Let $E(N)$ be the number of embedding parameters for a model with total size $N$ (in billions). Then the weight-memory budget in Gb is

$$M = 16 E(N) + P_w\big(N - E(N)\big) = P_w N + (16 - P_w) E(N). \tag{15}$$

For a fixed budget $M$ and a chosen precision $P_w$, the largest feasible parameter count $N$ is generally not free: it is implicitly defined by (15) ($E(N)$ is generally unknown and there is generally no clear rule for estimating it based on $N$). We denote this maximal feasible size by $N(P_w)$.

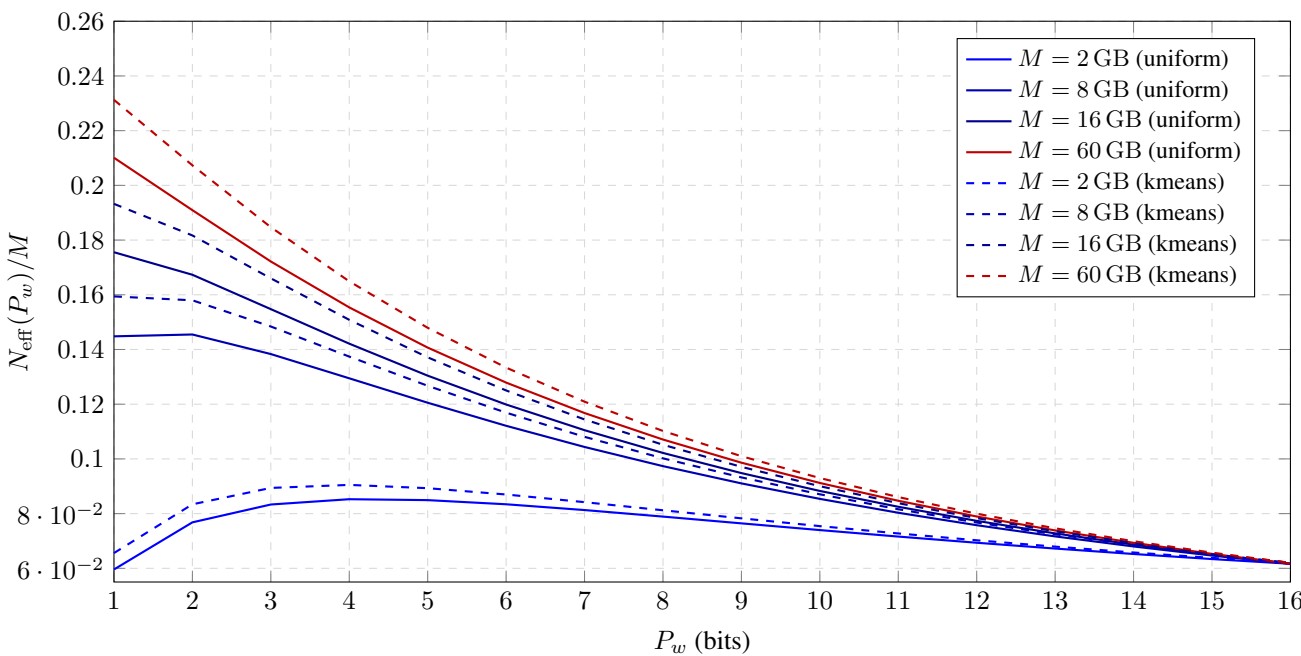

*Figure 7.* Effective parameter density $N_{\text{eff}}(P_w)/M$ versus backbone bit-width $P_w$ for fixed weight-memory budgets $M$ (GB). Solid lines: uniform quantization; dashed lines: kmeans quantization. The 16-bit special block consists of *both* token embeddings and the LM head ($E(N) = 2V d(N)$). For large budgets the density is maximized at the smallest feasible $P_w$, while at very small budgets the optimum can shift upward because the 16-bit special block becomes non-negligible. Interestingly, the optimal bit-width for uniform formats at 8 GB memory-budget is 2 bits, while the kmeans format yields 1 bit as the optimal bit-width.

**Objective: maximize effective capacity per memory.** Since $M$ is fixed, it is convenient to maximize effective capacity normalized by memory,

$$\frac{N_{\text{eff}}(P_w)}{M} = \frac{f(P_w)\,N(P_w)}{M}, \qquad \text{and we seek} \qquad P_w^\star \in \arg\max_{P_w} \frac{f(P_w)\,N(P_w)}{M}. \tag{16}$$

In general, there is no closed-form reduction because $E(N)$ depends on architecture and scales with $N$, so $N(P_w)$ must be obtained by solving (15). Practically, we evaluate a small set of supported precisions $\{P_w\}$; for each $P_w$ we solve (15) for $N(P_w)$, then compute (16), and select the maximizer. We therefore require an explicit mapping $N \mapsto E(N)$ to evaluate (15) and (16).

**Hidden-size scaling from observed model classes.** For Llama-style architectures, we fit a power law mapping total parameters $N$ to hidden size $d(N)$,

$$d(N) \approx d_0 \left(\frac{N}{N_0}\right)^\alpha, \qquad N_0 = 3\,883\,551\,744, \;\; d_0 = 3072, \;\; \alpha \approx 0.320. \tag{17}$$

The fit is anchored at the 3.88B class and matches the following observed pairs $(N, d)$ in log-space:

$$(3\,883\,551\,744,\; 3072), \;\; (11\,520\,053\,248,\; 4096), \;\; (30\,643\,279\,872,\; 6144).$$

(For implementation, $d$ is typically rounded to a hardware-friendly multiple; here we keep (17) continuous.)

**Embedding parameter count.** With vocabulary size $V = 128\,256$ and untied embedding and LM head matrices of shape $V \times d$, the embedding parameter count is

$$E(N) = 2V d(N) \approx 2 \cdot 128\,256 \cdot 3072 \cdot 10^{-9} \left(\frac{N}{3.88}\right)^{0.320}. \tag{18}$$

Substituting (18) into (15) yields the implicit relation defining $N(P_w)$ for each $P_w$ and fixed $M$, and thus the effective density (16).

### C.1. Regime-dependent optimum

We plot, for different memory budgets $M$ in GB (Gigabyte), the effective capacity per unit of memory as a function of the precision $P_w$ in Figure 7. As discussed in the previous subsections, for each possible bit-width $P_w$, we maximize the total parameter count $N$ such that the memory budget constraint in Equation (15) is satisfied. We then plot $N_{\text{eff}}(P_w)/M$.

## D. Theoretical model of software-decoded formats

We wish to build a basic theoretical model of speedup, based on a roofline hardware model of a device that supports $R_{\text{compute}}$ arithmetic operations (multiplies or adds) per second and can transfer $R_{\text{transfer}}$ bytes per second from its memory.

The fundamental operation is the matrix multiplication $Y^T = WX^T$ where $Y$ and $X$ are a bf16 matrices of shape $m \times h$, $m$ is a batch dimension, and $h$ is a hidden dimension. $W$ is a quantized parameter matrix of shape $h \times h$, with an average bits/parameter of $P_w$.

In many inference scenarios, we can assume that the activations $X$ and $Y$ can be retrieved from on-chip caches, so we need only consider the transfer of $W$ from memory. (We note that the results would not change substantially if this wasn't the case.) Therefore the time required for compute and parameter transfer are

$$t_{\text{compute}} = 2 \cdot m \cdot h^2 / R_{\text{compute}} \tag{19}$$

$$t_{\text{transfer}} = P_w \cdot h^2 / (8 \cdot R_{\text{transfer}}), \tag{20}$$

where the factor of 2 accounts for multiplies and adds separately, consistent with standard definitions of $R_{\text{compute}}$ op/s, and the factor of /8 accounts for the measurement of $P_w$ in bits whereas $R_{\text{transfer}}$ is commonly measured in bytes.

In our model, we assume ideal overlapping of communications and compute, in which case $t = \max(t_{\text{compute}}, t_{\text{transfer}})$. This gives

$$t = \frac{2 \cdot m \cdot h^2}{R_{\text{compute}}} \max\left(1, \frac{P_w \cdot R_{\text{compute}}}{16 \cdot R_{\text{transfer}}}\right). \tag{21}$$

If we set $\nu := R_{\text{compute}}/R_{\text{transfer}}$, and take the ratio $t_1/t_2$ to find the speedup associated with switching from $P_{w1}$ to $P_{w2}$ *with the same* $R_{\text{compute}}$, we obtain Equation (8).

**Regimes** As highlighted in Figure 3, we observe that Equation (8) predicts three performance regimes based on the batch size $m$.

1. For small $m < \min(P_{w1}, P_{w2}) \cdot \nu/16$, the speedup$_{1\rightarrow2}$ is the ratio of the precisions $P_{w1}/P_{w2}$. Both settings are memory-bound, so the time taken is proportional to the model size in bytes.

2. For intermediate $m$, the speedup scales $\propto 1/m$.

3. For large $m > \max(P_{w1}, P_{w2}) \cdot \nu/16$ the speedup is 1 (no speedup), as both settings are compute-bound, requiring approximately the same amount of compute work.

## E. Benchmarking

To demonstrate the feasibility of 4-bit and 1-bit nonlinear block-scaled weight formats, we evaluate the performance of optimized kernels for these formats, running on a contemporary inference-focused deep learning hardware accelerator. Our micro-benchmarks of a single kernel execution and whole-model benchmarks show strong speedups using 4 and 1-bit weights for a given model size at batch size 1 (for single-user autoregressive generation without speculative sampling), and some speedup at batch size 16, however no speedup by batch size 256.

Aside from the input token embedding lookup, which is computationally trivial, all matrix parameters in the Llama-like transformer architecture are consumed by token-wise projections, i.e. matrix multiplications where the local batch size $m$ is the product of the sequence length (during context "prefill" only) and "user" batch size. Therefore the fundamental operation for low-precision inference in the transformer architecture is the fused dequantize-multiply, $Y^T = \text{dequantize}(\tilde{W})X^T$, where $\tilde{W}$ is a tuple incorporating the quantized element data, block scales and nonlinear format centroids (if applicable).

We provide full kernel and benchmark code for sake of reproducibility.

### E.1. Kernel implementations

We explore multiple implementations of the fused dequantize-matrix-multiply operation. All are semantically similar to the following PyTorch code, which uses a 256-element (8-bit index) lookup table to map each packed input byte to $2 \times 4$-bit values or $8 \times 1$-bit values, followed by block scaling to reproduce the original weights.

```python
def dequant_matmul(x, Wq, Wscale, lut8):
    """
    x      :: batch_size x input_size x bf16
    Wq     :: output_size x (input_size//elements_per_byte) x uint8
    Wscale :: output_size x (input_size//block_size) x bf16
    lut8   :: 256 x elements_per_byte x bf16
    return :: batch_size x output_size x bf16
    """
    Wu = lut8[Wq.long()].flatten(start_dim=1)
    B = Wu.shape[1] // Wscale.shape[1]
    W = Wu.view(*Wscale.shape, B).mul(Wscale.unsqueeze(2)).flatten(start_dim=1)
    return x @ W.T
```

A minor variation is to use *deferred scaling*, where the block scaling is deferred until after the accumulation of unscaled weights across a single block:

```python
def dequant_matmul_defer_scale(x, Wq, Wscale, lut8):
    Wu = lut8[Wq.long()].flatten(start_dim=1)
    B = Wu.shape[1] // Wscale.shape[1]
    z = torch.einsum("mGg,hGg->mhG", x.unflatten(1, (-1, B)), Wu.unflatten(1, (-1, B)))
    return torch.einsum("mhG,hG->mh", z, Wscale)
```

Both methods produce similar results, with subtle differences since floating-point operations do not exactly follow distributivity/associativity. The original approach requires $2 \cdot m \cdot h^2 + h^2$ floating-point operations, while deferred scaling requires $2 \cdot m \cdot h^2 + 2 \cdot m \cdot h^2/B$ floating-point operations. Which is better in practice depends on hardware and compiler support.

We evaluate three implementations based on this strategy:

- Using `torch.compile(mode=max-autotune-no-cudagraphs)` on similar PyTorch code to the example shown above, with `bf16` compute.

- Custom Triton (Tillet et al., 2019) kernels for chunked matrix-vector and matrix-matrix. We rely on the Triton autotuner to select chunk shape hyperparameters from a set of useful configurations identified by a broad sweep on the evaluation GPU system. We also find empirically that in all preferred configurations it is beneficial to disable automatic pipelining with `num_stages=1`, and to use `num_warps=1` for most configurations. This method also uses `bf16` compute.

- An adaptation to the Marlin (Frantar et al., 2024) 4-bit linear matrix multiply with block scaling, where we add our 8-bit LUT and adapt the code for $B = 64$. This implementation only supports 4-bit formats and uses `float16` for activations and compute. We note that the GPU system evaluated supports `float16` at the same speed as `bf16`.

### E.2. Micro-benchmarks

For our micro-benchmarking evaluation, we test applicable implementations across a range of relevant $m$ (equivalent to batch size), $h$ (hidden dimension) and bit-widths $P_w$. We create a CUDA graph of 100 calls to the kernel, using different input and output tensors for each call (to avoid unrealistic caching advantages). All calls take place on the same stream, so there is no overlap between executions. We invoke this CUDA graph 100 times, reporting the average wall-clock time of a single kernel invocation and the standard error across runs.

Note that we report micro-benchmark results solely for square projections where $W$ is a $h \times h$ matrix. While some transformer model projections are commonly square (e.g. attention query and output projections), many are not (e.g. feedforward network up, gate and down projections). In early experiments, we observed that the key performance trends do not vary substantially for non-square projections, therefore we omit these results for brevity. Non-square projections are necessarily present in the model benchmarks of Section E.3.

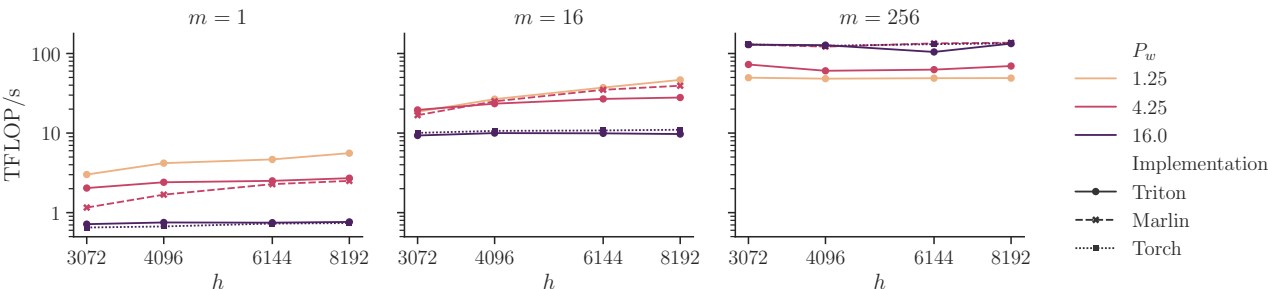

*Figure 8.* A micro-benchmark of kernel performance for square matrix multiplication, swept over batch size $m$, hidden size $h$ and bit-width. The maximum error bar ($\pm 2$ standard errors) is approximately $\pm 1\%$ of the mean, so error bars are not visible at this scale.

**Setup**  The test platform is an NVIDIA L40S GPU, which has $864\,\text{GB/s}$ memory bandwidth and peak dense matrix compute of 362 TFLOP/s in bf16. Applicable software versions are PyTorch 2.9.1, Triton 3.5.1 and CUDA 12.8.

**Results**  Headline results for batch size $m = 1$ and a fixed problem size are shown in Table 6. We expect good speedups in this case, since the kernel execution should be dominated by the time taken to transfer the weight matrix from memory. Effective bandwidth measures the operand sizes divided by the time taken, and can be used as an efficiency measure for such memory-bound kernels.

Both Triton and Marlin implementations are able to achieve good, almost optimal, speedups for 4-bit dequantization using a lookup table and block scaling, but the 1-bit Triton kernel is not able to saturate memory bandwidth in the same way, due to a smaller amount of data to transfer and more dequantization and compute work per byte transferred.

*Table 6.* A micro-benchmark of different kernel implementations, for $m = 1$ and $h = 8192$. The theoretical maximum speedup for 4-bit and 1-bit kernels is $3.8\times$ and $12.8\times$ respectively. Standard errors are $< 0.2\,\mu\text{s}$ except for the torch.compile implementation ($< 0.6\,\mu\text{s}$).

| $P_w$ | Implementation | Time (Speedup) | Effective Bandwidth |
|---|---|---|---|
| 16 | Torch | $181.6\,\mu\text{s}\ (1.0\times)$ | $739\,\text{GB/s}$ |
| 16 | Triton | $175.6\,\mu\text{s}\ (1.0\times)$ | $764\,\text{GB/s}$ |
| 4.25 | torch.compile | $772.3\,\mu\text{s}\ (0.2\times)$ | $46\,\text{GB/s}$ |
| 4.25 | Marlin | $53.5\,\mu\text{s}\ (3.4\times)$ | $667\,\text{GB/s}$ |
| 4.25 | Triton | $49.5\,\mu\text{s}\ (3.7\times)$ | $721\,\text{GB/s}$ |
| 1.25 | torch.compile | $770.4\,\mu\text{s}\ (0.2\times)$ | $14\,\text{GB/s}$ |
| 1.25 | Triton | $24.0\,\mu\text{s}\ (7.6\times)$ | $438\,\text{GB/s}$ |

A broader sweep of results over various $m$, $h$ and bit-widths is shown in Figure 8. This shows that both Marlin and Triton implementations retain a (smaller) speedup over full bf16 precision at batch size $m = 16$, but the advantage is lost by $m = 256$. This is consistent with the transition from the memory-bound small-batch regime to the compute-bound large-batch regime. The Marlin kernel is particularly strong at intermediate batch sizes. We note that a similar technique of custom CUDA combined with an optimized data layout would also be applicable to 1-bit dequantization, opening the possibility for further improvements.

### E.3. Model benchmarks

We also evaluate the same kernel implementations in the context of autoregressive token generation, which is a memory-bound operation at small batch size due to the lack of sequence parallelism. We use the Llama 3 model implementation from transformers (Wolf et al., 2019), with model configurations from Section 3.2, which retain the embedding and final projection matrices in bf16, quantizing all other matrix parameters, and capture the forward pass in a CUDA graph. We generate 100 tokens and record the average time per token. The hardware setup is as described in Section E.2.

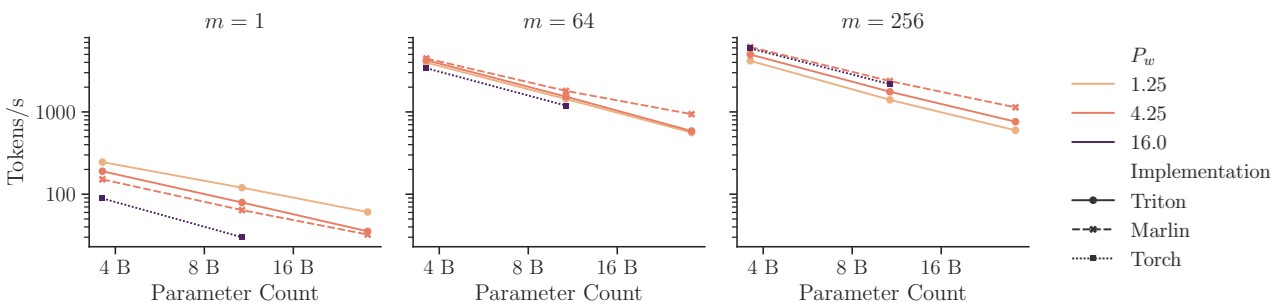

*Figure 9.* Autoregressive decoding speed for different model size, batch size and bit-width. Larger batches are beneficial to token/s, since they amortize the cost of transferring the model parameters, but this has greater impact on the `bf16` baseline than on compact formats. The maximum error bar ($\pm 2$ standard errors) is $< 0.5\%$ of the mean.

**Results**  Main results for batch size 1 are shown in Table 7, showing that 4 and 1-bit formats give consistent speedups for a given model size, in this memory-bound setting. Importantly, the 4-bit 12B model is only $\approx 10\%$ slower than the 16-bit 4B model, however the 1-bit 31B model is somewhat slower. We also observe the small overhead of nonlinear quantization in Marlin by comparing against the original *Marlin (Linear)* dequantization approach that uses arithmetic operations in place of a lookup table.

*Table 7.* Model tokens/s for autoregressive generation at batch size 1. Entries in blue correspond to our scaled-up training settings from Section 3.2 The 4-bit 12B model is $\approx 10\%$ slower than the 16-bit 4B model, while the 1-bit 31B model has somewhat lower tokens/s since the dequantization and higher-precision arithmetic is unable to saturate memory bandwidth (see Table 6). The maximum error bar ($\pm 2$ standard errors) is $< 1\%$ of the mean.

| $P_w$ | Implementation | 4B | 12B | 31B |
|---|---|---|---|---|
| 16 | Torch | 88.8 | 30.0 | OOM |
| 4.25 | Triton | 190.1 | 79.3 | 35.4 |
| 4.25 | Marlin | 151.5 | 64.1 | 32.4 |
| 4.25 | Marlin (Linear) | 156.2 | 65.5 | 32.8 |
| 1.25 | Triton | 245.6 | 120.0 | 60.7 |

Further results are shown in Figure 9. Consistent with micro-benchmarking outcomes, the advantage of weight quantization reduces at batch size 64 and disappears by batch size 256. This shows the applicable regime for formats without hardware support, which can accelerate execution during small batch token generation, but can instead impose some overhead at large batch sizes, including context prefill. In such settings, it may be more efficient to dequantize in a separate kernel, before using a standard high-precision GEMM kernel for the compute.

