# OpenReview forum: "1-Bit Wonder: Improving QAT Performance in the Low-Bit Regime through K-Means Quantization"
_ICML.cc/2026/Conference — ICML 2026 regular_

### Official Review · Reviewer_CEbB · 2026-03-08

**Soundness:** 2
**Presentation:** 3
**Significance:** 2
**Originality:** 2
**Overall Recommendation:** 3
**Confidence:** 4

**Summary:**

This paper studies the scaling behavior of vanilla quantization aware training (QAT) with a straight through estimator (STE) with scalar and vector quantizers. The paper shows that vector quantization outperforms scalar quantization in QAT settings, and that VQ is more stable than SQ for a fixed bitrate.

**Compliance With Llm Reviewing Policy:**

Affirmed.

**Final Justification:**

See my rebuttal acknowledgement. From my comment that the AC should be able to see (not sure if the authors can see): I think this paper could use another revision with a more scoped out experimental setting and more robust scaling laws. I will keep my score.

**Key Questions For Authors:**

See above.

**Limitations:**

See above.

**Strengths And Weaknesses:**

I am concerned with the novelty of this submission. It is well known that VQ can achieve lower distortion than SQ (there are hundreds, if not thousands of papers in the information theory literature showing this), and the rest of the paper just studies QAT scaling at  experiment scales. There does not seem to be a core technical contribution, and there are already many peer-reviewed papers out there studying the scaling behavior quantization aware training and quantization in general. If the authors had instead studied the behavior of QAT/D on open source LLMs that achieve actual frontier performance, perhaps their submission would have been more convincing or they would have observed different Pareto frontiers. This paper also mischaracterizes the state of PTQ. There are now hundreds of PTQ papers showing that PTQ works well far below 4bpw, specifically by using techniques such as vector quantization (AQLM, QuIP#, etc.).

---

> ### Author Rebuttal · Authors · 2026-03-27
>
> We thank the reviewer for their feedback. We kindly ask the reviewer to read our response, as there seems to be some misunderstanding regarding the scope of our work. We also hope to have addressed the concerns that the reviewer listed.
>
> > W1: “*The paper shows that vector quantization outperforms scalar quantization in QAT settings, and that VQ is more stable than SQ for a fixed bitrate.*”
>
> We seek to clarify, as this does not accurately reflect the work. We employ the k-means algorithm for nonlinear _scalar_ quantization, and our work compares nonlinear to linear (uniform) scalar quantization. This is explained in S1 _contributions_ (L099) and S2.2 _Nonlinear k-means quantization_ (L165). Note that we employ vector lookup tables only as a kernel design trick in S3.3; the format remains scalar and the lookup table is a cartesian product of the 1D centroids.
>
> > W2: “*I am concerned with the novelty of this submission. It is well known that VQ can achieve lower distortion than SQ (there are hundreds, if not thousands of papers in the information theory literature showing this)*”
>
> In light of the correction above, this concern does not apply to our paper.
>
> A similar argument could be applied to nonlinear versus uniform scalar quantization. However, our work is an empirical study of the scaling behaviour with respect to weight quantization when using QAT specifically, rather than a general claim regarding distortion, which is well-known.
>
> > W3: “*There does not seem to be a core technical contribution, and there are already many peer-reviewed papers out there studying the scaling behavior quantization aware training and quantization in general.*”
>
> Our contributions are described in S2 (L089). We advance upon existing scaling law studies “Scaling laws for Precision” [1] and ParetoQ [2] by incorporating block scaling and nonlinear formats. Nonlinear block-scaled formats are a generalisation of MXFP and NVFP formats which are important developments in low-precision numerics. We present a parameter-memory-matched downstream evaluation that we have not seen elsewhere.
>
> > W4: “*If the authors had instead studied the behavior of QAT/D on open source LLMs that achieve actual frontier performance, perhaps their submission would have been more convincing or they would have observed different Pareto frontiers.*”
>
> The precision-scaling behaviour of models under fine-tuning QAT/D is of great interest. Like the reviewer, we might expect to observe different optimal frontiers in this setting. We do not claim generality to this fine-tuning setting, but argue that our work considers a relevant question: “when training an LLM on a fixed memory budget, what is the optimal precision-parameters tradeoff?”
>
> > W5: “*This paper also mischaracterizes the state of PTQ. There are now hundreds of PTQ papers showing that PTQ works well far below 4bpw, specifically by using techniques such as vector quantization (AQLM, QuIP#, etc.).*”
>
> In S4 L418, we say:
>
> > Overall, while modern PTQ methods can remain competitive down to ∼4-bit weights, their performance degrades rapidly below this regime
>
> We understand our statement to hold for the papers you referenced. AQLM and QuIP# at 2-3 bits have considerably higher perplexity and worse task performance than the 16-bit baseline (Tables 1 & 2, [3]), e.g. 7B Llama, 3-bit Wiki2 ppl 5.12 (FP16) -> 5.46 (3-bit AQLM). Many PTQ techniques (unfortunately not AQLM and QuIP#) are compared against QAT in ParetoQ [2], Tables 3-5, which show that while downstream performance is close for 4-bit formats, the gap between PTQ and QAT widens at 3-bit and 2-bit.
>
> We propose clarifying our statement: “their performance degrades rapidly relative to QAT…”, and hope that this would address your concern.
>
> ---
>
> [1] Scaling laws for Precision, Kumar et al., 2024.
>
> [2] ParetoQ: Improving Scaling Laws in Extremely Low-bit LLM Quantization, Liu et al., 2025.
>
> [3] Extreme Compression of Large Language Models via Additive Quantization, Egiazarian et al., 2024.
>
>
> ---
>
> We would be happy to discuss any further questions the reviewer might have. In light of our clarification and responses, we would be grateful if the reviewer would consider updating their score.

---

> > ### Author Rebuttal · Reviewer_CEbB · 2026-04-01
> >
> > Thanks for your response. It appears that I misunderstood how K-Means was being used here and what the quantizer setup was. That said, I am still not fully satisfied with the technical novelty or real-world practicality of this submission.
> >
> > 1. The main contribution seems to be empirical scaling laws based on performing QAT with uniform and lloyd-max scalar quantizers. The QAT setup is to perform high precision training for 1000 steps (3% of the training run), and then to switch over to weight-only QAT with a STE on the weights. This is not really a realistic setting. In practice, QAT is performed after full high-precision pretraining and only done at the end for ~1% of the token budget (see Gemma 3, the experimental setup in ParetoQ). What this paper has instead effectively studied is low-precision weight-only pretraining (+ some SFT at the end), which is closer to what BitNet does. Unfortunately, BitNet-style training hasn't been shown to be practical over regular HP training + QAT, and this paper doesn't address that problem. People who do low-precision pretraining quantize both the activations (which are much harder to quantize than the weights!) and weights to hardware supported datatypes to get wall-clock speedups. Quantizing the weights alone gives you none of that and hurts quality over the HP+QAT alternative, which doeesn't cost any more.
> >
> > 2. The scaling laws don't take into account any of the properties of quantizer centroids or weight distribution. The only property of the K-means quantizers used in scaling law fitting is the effective weight precision $P_w$ in $N_{eff}$, and the K-means quantizers are only differentiated from the uniform quantizers through fitting separate sets of coefficients. Ideally, a good scaling law would be able to take properties of the quantizer itself into account so that manual separation of K-means and uniform quantizers isn't needed.
> >
> > 3. The K-means quantizers are initialized once at the beginning of the QAT stage and used for the rest of training. The explanation given for this in the paper is that this stabilizes training and allows the QAT process to adapt to the weights. Can you provide evidence of this? For example, the entropy of the quantized weights should remain stable and high if this is true. I have a hard time believing that the weight distribution does not change significantly past the first 3% of the training run.
> >
> > 4. PTQ: ParetoQ doesn't do much better than the PTQ baselines, if at all. I haven't personally seen much evidence that open-source QAT methods do much better than PTQ on things like KL divergence and perplexity. This is also why I think your submission would be strengthened if it studied the scaling laws of various weight formats in a realistic QAT setting, like QAT after high precision pretraining.
> >
> > I will raise my score to a 3. My main remaining reservations are (1) the realisticness of the QAT setting studied here and (2) how useful the scaling laws actually are.

---

> > > ### Author Response · Authors · 2026-04-06
> > >
> > > Thank you for your further questions and outlining your outstanding concerns.
> > >
> > > > W2-1: The main contribution... This is not really a realistic setting.
> > >
> > > While we agree that both quantization-aware fine-tuning (high-precision training followed by short QAT with/without distillation) and low-precision training (weights + activations for training speedup) are highly relevant settings that benefit from further study, we do not understand the BitNet-style QAT-from-scratch to have been conclusively shown to be unrealistic or impractical.
> > >
> > > As outlined in our response to reviewer Ebyh, our goal is to compare formats and bit-widths. Considering the results of ParetoQ [1], Figure 2, we might expect a moderate but consistent improvement if we delayed QAT until 90% training progress, assuming their results generalize across model size and dataset. It would be interesting to confirm this, but the possible improvement does not invalidate our setup, since ParetoQ does not claim that the bit-width tradeoff depends on FP/QAT timing.
> > >
> > > > W2-2: The scaling laws don't take into account any of the properties of quantizer centroids or weight distribution…
> > >
> > > This is an interesting suggestion, and one we would be very pleased to see realized. However, the set of possible formats is vast, encompassing centroid choice, block size, etc, and it is not obvious what properties of a quantizer would best abstract the many differences between formats. Since our compute budget did not permit a thorough evaluation of multiple formats, we chose two strong and popular options. Without a wider sweep of formats, we could not claim a more general law, so the fairest presentation of our findings was two sets of fitted coefficients.
> > >
> > > > W2-3: The K-means quantizers are initialized once at the beginning of the QAT stage…
> > >
> > > This is a helpful point. Our early experiments identified the recipe we presented as being stable and effective, however we do not have comparable results to ablate these design choices, so cannot robustly make the case that this is the optimal recipe. We would emphasise that any improvements here would only strengthen nonlinear formats, so our main finding that they present a substantial benefit would hold.
> > >
> > > _(Edited)_ Regarding shifts in the marginal distributions of weights during training, we have generated weight histograms from intermediate checkpoints of a 2-bit model in [1bw_weight_distributions.pdf](https://57409622.s3.eu-central-1.amazonaws.com/1bw_weight_distributions.pdf). These show scale growth during training, which is captured by the dynamic block scaling factor (recalculated whenever the master weights are updated). Visually, the shape seems relatively stable after 2000 steps, but we don't rule out that the shape could vary enough to benefit from also retraining the k-means centroids, leading to a further improvement for the nonlinear format over uniform.
> > >
> > > > W2-4: PTQ: ParetoQ doesn't do much better than the PTQ baselines, if at all…
> > >
> > > Our understanding differs regarding ParetoQ - in Tables 3-5, their performance seems better than the presented PTQ baselines, on downstream task averages. The WikiText2 perplexity comparison seems to vary, depending on model and bit width. For example, the 3-bit results of Table 4 show better average downstream accuracy across the board, best perplexity for Llama, with worse perplexity only for the MobileLLM family. In our experience of the fine-tuning setting, QAT with cross-entropy or KL divergence distillation loss performs very well on the training / fine-tuning data used (which is unsurprising) - the challenge can be finding a datamix that gives good generalisation. PTQ methods may have some advantage in this regard, as despite using some “calibration data”, their performance may depend less on the data distribution than QAT. However both lines of research are promising.
> > >
> > > ---
> > >
> > > We are grateful for the increase in score, and as a headline response to your reservations,
> > >
> > > 1. _the realisticness of the QAT setting studied here_. First, we argue the QAT-from-scratch setting has not been shown to be sub-optimal across different formats and training settings (we are only aware of ParetoQ Figure 2, with a single format family). Second, we note that ParetoQ indicates that full-precision training for 90% of steps is an opportunity for improved performance in general, but may not change the relative performance of different formats, meaning our conclusion regarding very-low-bit nonlinear formats would stand.
> > >
> > > 2. _how useful the scaling laws actually are_. Ours are, to the best of our knowledge, the first quantization scaling laws for modern block-scaled formats, and address two of the most popular element formats: uniform (INT), and nonlinear (k-means). While there is substantial scope for further work, we argue that these are useful for the community to build upon.
> > >
> > > ---
> > >
> > > [1] ParetoQ: Improving Scaling Laws in Extremely Low-bit LLM Quantization, Liu et al., 2025.

---

### Official Review · Reviewer_Ebyh · 2026-03-12

**Soundness:** 3
**Presentation:** 3
**Significance:** 3
**Originality:** 2
**Overall Recommendation:** 4
**Confidence:** 4

**Summary:**

This work extends scaling law to answer question of allocating parameter count and weight precision under fixed memory budget. From the analysis, authors found that scaling law prefers more parameters than more bits per parameter. Together, authors show k-means based weight quantization outperforms uniform integer quantization in quantization-aware training (QAT). Experimental results are shown with Nemotron-CC with varying precision and model size with matching budget.

**Compliance With Llm Reviewing Policy:**

Affirmed.

**Final Justification:**

While some concerns remain, I believe this work has merit for acceptance. Accordingly, I am increasing my score.

**Key Questions For Authors:**

+ Could authors share insights about applicability of extended scaling law to variety of models and effect of FP training stage?
+ Although the paper provides valuable insights, k-means based quantization itself is not novel and its effectiveness over uniform quantization has been studied in prior work. Thus the overall contribution may appear limited. Could the authors clarify how this affects the overall contribution of the paper?

**Limitations:**

Authors discussed limitations in the paper with independent section

**Strengths And Weaknesses:**

[Strengths]
+ Authors extend scaling laws for precision to provide valuable insights for the optimal allocation of precision and number of parameters under a given buget.
+ The paper provides theoretical and practical analysis on efficiency of k-means based quantization.
+ Paper is well-organized and easy to follow

[Weaknesses]
+ The extended scaling law is examined only for a single family of models.
+ As mentioned by authors and prior work, timing of introducing QAT has been reported to be importatnt for the performance. However its effect is not analyzed or incorporated into scaling law.
+ While authors claim design space of quantization has not been fully explored and demonstrate effectiveness and efficiency of k-means based quantization as an example, the approach itself is not novel at all. As a form of non-uniform quantization, k-means based quantization and its acceleration have been studied in various prior works. [1], [2]

[1] DKM: Differentiable k-Means Clustering Layer for Neural Network Compression (ICLR’22)
[2] KLLM: Fast LLM Inference with K-Means Quantization (Arxiv)

---

> ### Author Rebuttal · Authors · 2026-03-27
>
> We thank the reviewer for their honest feedback and for their appreciation of the overall message and strengths. We seek to address the main concerns regarding novelty and transferability of our findings across models and QAT recipes below.
>
> > Q1: “*Could authors share insights about applicability of extended scaling law to variety of models and effect of FP training stage?*”
>
> It would be interesting to investigate how this scaling law might vary based on model architecture, however, consistent with standard practice, we use a specific but highly relevant modern autoregressive transformer architecture for our scaling evaluation. The Llama3 model architecture is well established and deviations from it in newer model families like Qwen3 are minimal. There is no concrete reason to believe that results will look different for those model families.
>
> We are also not claiming that our results are universal for every architecture. It could be the case that model components such as mixture-of-experts, local attention layers or state space layers are affected differently by quantization and could change the scaling law exponents. If the reviewer feels strongly about this aspect, we are happy to include it in our limitations section.
>
> Regarding the FP training stage, we agree with the reviewer that the placement of QAT into an end-to-end model training pipeline is an important consideration. However, in this work we explicitly only focus on comparing formats and different bit-widths. We therefore chose a fixed setup that we believe is suitable for comparison. Our recipe introduces quantization after 1K (of 12K total) steps, i.e. at 8% training progress. Considering the results of ParetoQ [3], Figure 2, we might expect a moderate but consistent improvement if we delayed QAT until 90% progress, assuming their results generalize across model size and dataset. It would be interesting to confirm this, but the possible improvement does not invalidate our setup, since ParetoQ does not claim that the bit-width tradeoff depends on FP/QAT timing.
>
> > Q2: “*Although the paper provides valuable insights, k-means based quantization itself is not novel and its effectiveness over uniform quantization has been studied in prior work. Thus the overall contribution may appear limited. Could the authors clarify how this affects the overall contribution of the paper?*”
>
> > “*As a form of non-uniform quantization, k-means based quantization and its acceleration have been studied in various prior works. [1], [2]*”
>
> We agree that utilizing a k-means quantizer in deep learning is not novel. Instead of offering a novel quantization technique, our work seeks to combine modern QAT techniques, offering the conclusions regarding the efficacy of low-bit formats as our main contribution. We hope this is accurately reflected in our contributions section, and are pleased to update it if helpful.
>
> We are grateful for the two references that the reviewer mentions, and will update our related work section to discuss these. In the following we highlight some key differences in scope and results between the mentioned papers and our work.
>
> [1]  Here, the authors focus on improving k-means clustering by relaxing the rounding step with a softmax approximation. The experimental section is very different from ours in scope, as they evaluate on vision models and non-generative NLP tasks with a Bert model. It is unclear if their method scales to modern LLMs and would outperform our more vanilla k-means method.
>
> [2] This work mostly focuses on efficient k-means inference. For their experiments they use PTQ, which makes the setting fundamentally different from ours. Also, notably they don’t go below 4 bits for the weight precision (which makes sense given the PTQ setting), where our best model is in 1-bit precision. Finally, they only report logprob based eval scores, in contrast to our more comprehensive evaluation on generative downstream benchmarks. Therefore our work is complementary to this.
>
> ---
> [1] DKM: Differentiable k-Means Clustering Layer for Neural Network Compression (ICLR’22)
>
> [2] KLLM: Fast LLM Inference with K-Means Quantization (Arxiv)
>
> [3] ParetoQ: Improving Scaling Laws in Extremely Low-bit LLM Quantization (NeurIPS’25)
>
> ---
>
> **Summary**: We strongly believe that our main contribution provides improved understanding on the trade-off between bit-width and parameter count with QAT. In particular, that we have met the originality standard from the ICML’26 reviewer guidelines, which state:
>
> > Originality does not necessarily require introducing an entirely new method. Rather, a work that provides novel insights by evaluating existing methods, or demonstrates improved understanding is also equally valuable.
>
> In light of our responses, we ask if the reviewer might consider updating their score.

---

> > ### Author Rebuttal · Reviewer_Ebyh · 2026-04-03
> >
> > Despite some concerns, I consider this work to have merit for acceptance. I am therefore increasing my score.

---

### Official Review · Reviewer_Z6N6 · 2026-03-13

**Soundness:** 3
**Presentation:** 4
**Significance:** 3
**Originality:** 3
**Overall Recommendation:** 5
**Confidence:** 3

**Summary:**

This paper presents quantization scaling laws for LLMs, illustrating that QAT with few-bit LLMs achieve very competitive model quality and performance under fixed memory budgets. The authors present k-means quantization as a superior method as compared to uniform quantization in their QAT schemes.

The main weaknesses lie in the overall search space. The claim that kmeans quantization beats uniform quantization may only be valid in this limited setting: weight-only quantization, no PTQ, no outlier mitigation channels, 1-bit and 4-bit settings. It would greatly strengthen this paper if the authors were to expand to consider PTQ technique. Furthermore, discussion on dataset size is limited and it would be helpful to provide additional figures illustrating this contribution to final downstream accuracy.

**Compliance With Llm Reviewing Policy:**

Affirmed.

**Final Justification:**

The authors have addressed my rebuttal questions accurately, I maintain my score.

**Key Questions For Authors:**

Can the authors provide a clear comparison with ParetoQ? To strengthen the credibility of these model quality performance results, it would be helpful to understand in comparison to existing state-of-the-art results.

One of the most important confounding variables in quantization technique, such as a specific mapping using PTQ or advanced QAT methods. Can the authors provide additional results for how these figures may change over those settings?

The result that kmeans quantization beats uniform quantization is in an extremely limited context -- where no PTQ is applied and under 4-bit or 1-bit weight quantization. Have the authors considered activation quantization and could the authors provide results for additional quantization targets and mixed precision?

**Limitations:**

yes

**Strengths And Weaknesses:**

This paper has a technically solid foundation with a principled view of uniform and kmeans quantization. The scaling law presented has a clear purpose to understand how QAT settings influence overall model quality under a fixed memory budget. The paper also offers microbenchmarking results and memory bandwidth comparisons for various bitwidths.

---

> ### Author Rebuttal · Authors · 2026-03-27
>
> We thank the reviewer for the positive evaluation of our work and are happy to answer the raised questions.
>
> > Q1: “*Can the authors provide a clear comparison with ParetoQ? To strengthen the credibility of these model quality performance results, it would be helpful to understand in comparison to existing state-of-the-art results.*”
>
> Thank you for this helpful point. We agree that this would be useful for readers in general and will update the paper with a specific comparison. In summary: ParetoQ [1] utilizes uniform quantization formats, whereas we compare uniform and k-means quantization. We also utilize block scaling, whereas ParetoQ uses channel scaling. In their setting, the findings differ from ours (their Pareto-optimal bit-with is around 2). They also only train models up to 8B parameters compared to our 31B 1-bit model, and they only evaluate on logprob evaluations whereas we demonstrate the optimality of aggressive 1-bit quantization on generative downstream tasks.
>
> > Q2: “*One of the most important confounding variables in quantization technique, such as a specific mapping using PTQ or advanced QAT methods. Can the authors provide additional results for how these figures may change over those settings?*”
>
> With regards to our scaling law evaluation, we agree that varying quantization technique is likely to change the optimal precision for a given memory budget. In particular, PTQ techniques that typically incur higher degradation than QAT would likely prefer wider formats. Regarding advanced QAT methods, since improved quantization techniques would boost narrower formats more than wider ones (since narrower formats have more room for improvement), an improved recipe should be expected to result in an even stronger preference for narrow bitwidths.
>
> > Q3: “*The result that kmeans quantization beats uniform quantization is in an extremely limited context -- where no PTQ is applied and under 4-bit or 1-bit weight quantization. Have the authors considered activation quantization and could the authors provide results for additional quantization targets and mixed precision?*”
>
> With regards to PTQ, we agree that optimal choice of formats may be different in this setting. Since our investigation considers QAT as an alternative to PTQ which typically outperforms it in the low-bit regime (see ParetoQ, Table 3), we propose that this limitation does not diminish the relevance of our work.
>
> Regarding additional precision targets, we note that our scaling law used a fine-grained grid of precisions, e.g. {1.25, 2.25, 3.25, 4.25, 6.25, 8.25}-bit for k-means. Due to compute constraints, we employed a coarser grid for our scale-up experiments.
>
> Finally, we exclude activation quantization from our experiments since activation memory usage is a less pressing constraint during autoregressive inference. Thus, incorporating activation quantization would be an interesting extension, but we considered it out of scope for this work.
>
> [1] ParetoQ: Improving Scaling Laws in Extremely Low-bit LLM Quantization, Liu et al., 2025.

---

> > ### Author Rebuttal · Reviewer_Z6N6 · 2026-04-03
> >
> > Thank you, my questions have been resolved.

---

### Official Review · Reviewer_jmCS · 2026-03-13

**Soundness:** 3
**Presentation:** 3
**Significance:** 3
**Originality:** 3
**Overall Recommendation:** 4
**Confidence:** 4

**Summary:**

The paper investigates the trade-off between model scale and weight precision under a fixed inference memory budget, specifically focusing on extreme low-bit QAT settings. The authors conduct a precision-aware scaling law analysis to compare uniform integer quantizer against nonlinear k-means quantizer. They hypothesize that under a fixed memory budget, it is optimal to push for the lowest stable precision (1-bit) and reinvest the saved memory into scaling up the parameter count. To empirically validate this, they train three memory-matched models: a 4B parameter 16-bit model, a 12B 4-bit model, and a 31B 1-bit model. Their downstream generative evaluations show that the 31B 1-bit model generally outperforms the others.

**Compliance With Llm Reviewing Policy:**

Affirmed.

**Final Justification:**

The author has addressed some of my concerns, but the issue of long-context reasoning remains unresolved. I disagree with the claim that using AidanBench to assess multi-turn dialogue capabilities is equivalent to evaluating long-context reasoning, as multi-turn dialogues do not necessarily imply long contexts; at minimum, standard benchmarks such as LongBench should be included, and benchmarks like SWE-Bench that evaluate real-world problem-solving are neither tested nor discussed. Additionally, I agree with reviewer CEbB’s point in the weaknesses section that the paper’s main contribution does not appear to be grounded in a realistic setting, and further discussion would strengthen the submission. As a result, I have raised my rating to 4, but would classify it as a “borderline” rating.

**Key Questions For Authors:**

Please refer to the weaknesses.

**Limitations:**

Yes

**Strengths And Weaknesses:**

Strengths:

- Extending Chinchilla-style scaling laws to include a saturation function for bit-width is a clever and useful formalization. It provides a principled framework for navigating the model size vs. precision trade-off.
- The authors train models up to 31B parameters from scratch, which requires substantial compute and engineering effort. Testing on downstream generative tasks (e.g., HumanEval, GSM8K, MMLU) rather than just reporting validation loss adds significant credibility to their claims.

Weaknesses:

- The methodology relies on training models from scratch on a relatively tiny budget of <200B tokens. Modern LLMs of this scale (12B to 31B) are typically trained on tens of trillions of tokens. While this setup allows the authors to plot a localized scaling law, it severely undermines the practical applicability of the results. In practice, QAT is overwhelmingly applied to already pre-trained dense weights. It is highly possible that the observed superiority of the k-means quantizer is merely an artifact of this low-data, train-from-scratch environment (e.g., the linear quantizer might simply require more tokens to converge optimally). Because the authors ignore the standard paradigm of applying QAT to pre-trained weights, the core conclusions might be highly questionable. Besides, I notice the quantization parameters (e.g., scaling factor) of linear quantizer are not learnable, where it could be sub-optimal for a linear quantizer.

- The fact that a non-linear quantizer yields lower reconstruction error than a uniform linear quantizer is a well-known reality. The reason the industry defaults to uniform quantization is due to the massive computational overhead associated with non-linear formats. However, the authors fail to provide sufficient empirical data on training time, training GPU memory consumption, or training throughput against the uniform quantizer. Without discussing the practical costs of k-means QAT, the comparison is incomplete. Furthermore, the paper's information density is noticeably low; for instance, the "Related Work" section spends an excessive amount of text detailing PTQ methods, which are largely irrelevant to a paper proposing from-scratch QAT.

- The authors select 1D k-means as their non-linear quantizer but fail to provide sufficient motivation for why this specific method is optimal. The design space for non-linear quantization is vast (e.g., logarithmic). The paper lacks any comparative baseline or theoretical justification demonstrating that k-means is uniquely suited for this task compared to other non-linear alternatives.

- Extreme low-bit quantization is notorious for causing performance drop, which catastrophically degrades a model's ability to process long contexts. The authors only evaluate on standard short-context benchmarks (MMLU, GSM8K, etc.). To prove that a 31B 1-bit model is actually a viable replacement for a 16-bit model, it is crucial to evaluate its performance on long-context reasoning tasks and complex tool-use scenarios. Without these evaluations, we cannot know if the 1-bit model has fundamentally broken long-context attention mechanics at scale.

---

> ### Author Rebuttal · Authors · 2026-03-27
>
> We thank the reviewer for their comprehensive feedback. In particular we are happy that the reviewer values our evaluation on generative downstream tasks, which is often overlooked in LLM quantization literature.
>
> In the following we address the concerns listed and hope that the reviewer reconsiders the overall score of our submission.
>
> > W1: *"The methodology relies on training models from scratch on a relatively tiny budget of <200B tokens…"*
>
> We agree with the reviewer that our experimental setup is not directly applicable to frontier-scale LLM training, where QAT is typically incorporated into mid-training or post-training. However, for the scope of this work, we argue our setup is appropriate and necessary for a fair comparison.
>
> Our goal was to compare quantization formats and bit-widths under QAT. Since there is no canonical way of incorporating QAT into an end-to-end model pipeline (i.e. when and for how many tokens to do QAT), we chose a setting that, in our opinion, introduces the least amount of bias towards a particular format by using QAT throughout the pipeline. Optimizing over training pipelines (which might even have a different answer for different formats) is an important line of work but out of scope for us.
>
> We also disagree that our token budget is too small. We are not aware of any comparable work that uses significantly higher budgets. We are sure that the reviewer is well aware that compute for research papers like this is usually very limited.
>
> The reviewer also notes that the scaling factor of the linear quantizer is not learnable. The same is true for the k-means quantizer. The scale of both formats is “dynamic”, set at each step to the absmax or absmean of the block which is a standard approach. For testing learnable scales, the proper experiment would be to use a learnable scale for both formats, but this was out of scope for this work because of our limited compute resources.
>
> > W2: *"The fact that a non-linear quantizer yields lower reconstruction error than a uniform linear quantizer is a well-known reality…"*
>
> We disagree that k-means-QAT necessarily incurs "massive computational overhead". During training, the additional index look-up is negligible. Memory usage is also essentially unchanged, since the centroid table is tiny relative to the model weights and optimizer state.
>
> Consistent with this, our runs did not show any dramatic overhead in training-throughput or memory consumption (these measurements come from the same GPU type, but exhibit some variance due to different node allocations):
>
> **1-bit training**
> | Format | Avg TPS | Mem (GiB) |
> | -- | --: | --: |
> | Uniform | 14,170 | 51.77 |
> | K-means | 13,094 | 51.77 |
>
> **2-bit training**
> | Format | Avg TPS | Mem (GiB) |
> | -- | --: | --: |
> | Uniform | 12,596 | 51.77 |
> | K-means | 12,647 | 51.77 |
>
> Concerning inference, we note the comparison offered in Table 7, where the runtime gap between linear and nonlinear dequantization implemented in the Marlin kernel is small.
>
> > W3: *"The authors select 1D k-means as their non-linear quantizer but fail to provide sufficient motivation…"*
>
> In Section 2 we give a clear motivation for the choice of the k-means quantizer. We mention that quantization via rounding is optimal in terms of $L^2$ reconstruction error (Panter & Dite 1951) and that the optimal set of centroids is obtained via k-means algorithm (Lloyd 1982).
>
> > W4: *"Extreme low-bit quantization is notorious for causing performance drop..."*
>
> We agree that long-context robustness is an important concern for low-bit quantization. In fact, this is exactly one of our contributions we aim to make: unlike prior 1-bit literature, which reports only loss- or log-probability-based evaluations, we evaluate downstream generative evaluations.
>
> While further evaluation on long-context reasoning would be of interest, the current results already provide meaningful evidence that the 31B 1-bit model has not catastrophically broken long-horizon reasoning or conversational consistency.
>
> First, the model achieves the best result on AidanBench. Since AidanBench is specifically designed to evaluate consistency in multi-turn conversation, it directly addresses the reviewer’s concern about long-context interaction quality.
>
> Second, our existing reasoning and code evaluations already involve long successful generations. For example, on MATH500, among 500 correct responses, the average answer length is 450 characters and the maximum is 1525. On HumanEval Instruct, the average answer length is 1088 characters and the maximum is 2408. These are not consistent with a model whose long-context attention mechanics have collapsed.
>
> We accept that we do not yet include a dedicated suite of long-context tool-use benchmarks, and we will clarify this as a limitation. But we respectfully disagree that “we cannot know” whether the 1-bit model is viable beyond short-context tasks.

---

> > ### Author Rebuttal · Reviewer_jmCS · 2026-04-07
> >
> > Thanks for the response. The author has addressed some of my concerns, but the remaining issue regarding long-context reasoning has not been resolved.
> >
> > - First, I disagree with the author's claim that using AidanBench to assess multi-turn dialogue capabilities is equivalent to evaluating long-context reasoning. This is because multi-turn dialogues do not necessarily imply long contexts. Even if the author does not evaluate complex tasks, at the very least, an assessment should be conducted on standard benchmarks like LongBench. Furthermore, benchmarks such as SWE-Bench, which observe whether a model can solve real-world problems, have not been tested or discussed.
> >
> > - Additionally, based on the first comment in the "weaknesses" of my review, I agree with reviewer CEbB’s viewpoint that the main contribution of this paper seems not to be based on a realistic setting. I think this submission could be further strengthened if the authors can give more discussion on this.
> >
> > Therefore, I have decided to raise my rating to 4, but I would classify it as a 'borderline' rating.

---

> > > ### Author Response · Authors · 2026-04-08
> > >
> > > We are grateful for the feedback and for the increase in score.
> > >
> > > > W4+ First, I disagree with the author's claim that using AidanBench to assess multi-turn dialogue capabilities is equivalent to evaluating long-context reasoning. This is because multi-turn dialogues do not necessarily imply long contexts. Even if the author does not evaluate complex tasks, at the very least, an assessment should be conducted on standard benchmarks like LongBench. Furthermore, benchmarks such as SWE-Bench, which observe whether a model can solve real-world problems, have not been tested or discussed.
> > >
> > > We agree that further evaluations on established long-context benchmarks (including LongBench and SWE-Bench) would strengthen the empirical coverage of our work. In our initial response, we provided some evidence that the model can retain consistency over growing sequences, although we acknowledge that this does not substitute for the benchmarks mentioned. Note that our setting does not involve KV-cache compression or other mechanisms that specifically target context representation. As a result, there is no mechanism in our method that would be expected to specifically degrade long contexts beyond the overall quantization effects already captured in perplexity and downstream evaluations. In our setting, any general degradation due to low-precision weights is offset by increased parameter count, such that overall performance is improved. That said, we agree that explicit evaluation on benchmarks such as LongBench and SWE-Bench would further strengthen the paper, and we will include this as a limitation and direction for future work.
> > >
> > > > W1+ Additionally, based on the first comment in the "weaknesses" of my review, I agree with reviewer CEbB’s viewpoint that the main contribution of this paper seems not to be based on a realistic setting. I think this submission could be further strengthened if the authors can give more discussion on this.
> > >
> > > Thank you for this suggestion - as you note, this has been raised by multiple reviewers, and we would be pleased to update the paper with our argument for the applicability of our results, alongside the limitation that they have not been directly confirmed in a late-quantization setting. As discussed in our response to reviewer CEbB, we argue for the applicability of our main results regarding format selection and very-low-precision. In outline, we argue first that while there is evidence that QAT-from-scratch is suboptimal, it remains a reasonable evaluation setup. Second, that our results may be expected to generalize from the early-training recipe evaluated to a late-training recipe, noting that ParetoQ Figure 2, which demonstrates the benefits of late-training QAT, offers a consistent improvement across the formats studied in that work.

---

### Decision · Program_Chairs · 2026-04-30

**Decision:**

Accept (regular)

**Comment:**

The paper studies the trade-off between model scale and weight precision under a fixed inference memory budget, and provides an empirical scaling-law perspective on how parameter count and bit-width should be allocated in extreme low-bit QAT. Reviewers acknowledged the paper’s solid experimental effort, clear presentation, and potentially useful insights for the community, particularly the memory-matched comparisons across substantially different model scales and precisions. The rebuttal addressed several concerns, and two reviewers explicitly indicated that their main questions were resolved, while another reviewer raised their score and viewed the paper as borderline but ultimately on the positive side.

At the same time, the discussion highlighted important limitations. In particular, multiple reviewers questioned how well the studied setting reflects the more common practical pipeline of applying QAT after high-precision pretraining, and whether the conclusions would transfer directly to that setting. There also remains concern that the empirical coverage is incomplete, especially for long-context and more realistic downstream evaluations. In addition, the paper should be careful not to overstate novelty, since the main contribution is not a new quantization method but rather an empirical study that provides insight into precision–scale trade-offs under the chosen setup.

Overall, I find that the paper offers sufficient empirical value and a useful perspective to merit acceptance, although only narrowly. For the final version, the authors should more explicitly scope their claims to the specific training setting studied here, strengthen the discussion of practical applicability relative to late-stage QAT, clarify the paper’s relationship to prior work on k-means and low-bit quantization, and clearly acknowledge the lack of dedicated long-context evaluation as a limitation.